# Spatially and Temporally Guided Bayesian Optimization for Brain Effective Connectivity Learning from fMRI and EEG Data

## Abstract

Brain effective connectivity (EC) characterizes the causal and directional interactions among brain regions and plays a central role in understanding cognition and neurological disorders. Constructing EC networks from multimodal neuroimaging such as functional Magnetic Resonance Imaging (fMRI) and electroencephalography (EEG) is challenging, since most existing methods rely on feature concatenation or linear mapping, neglecting structural consistency and nonlinear cross-modal dynamics. In this work, we propose STBO-EC, a spatially and temporally guided framework for multimodal EC learning. First, we develop an anatomy-informed spatial alignment strategy that leverages known brain region coordinates to establish structurally consistent correspondences between EEG electrodes and fMRI regions. Second, we design a time-slice-based alignment and fusion mechanism to effectively bridge the temporal resolution gap between fast EEG activity and slow fMRI signals. Finally, to tackle the high dimensionality and nonlinear dependencies of fused multimodal data, we employ a low-rank parameterized Bayesian optimization method (DrBO), which enables efficient exploration of the exponential EC search space while providing uncertainty-aware inference. Experiments on two real EEG–fMRI datasets demonstrate that STBO-EC consistently outperforms state-of-the-art baselines across multiple evaluation metrics. The code is available at `https://anonymous.4open.science/r/STBO-EC-6D03`.

## 1 Introduction

Brain effective connectivity (EC) characterizes the causal and directional interactions among brain regions and is fundamental for understanding the neural basis of cognition, behavior, and neurological disorders Mamoon et al. (2025). Traditional statistical approaches, such as structural equation modeling (SEM) McIntosh (1994), dynamic causal modeling (DCM) Friston et al. (2003), and Granger causality analysis (GCA) Barnett & Seth (2014); Seth et al. (2015), have been widely applied for EC learning. However, these methods depend strongly on assumptions of linearity and stationarity, which makes them inadequate for capturing the nonlinear and dynamic characteristics of real brain networks Marinazzo et al. (2011); Stramaglia et al. (2014).

With the advancement of machine learning, data-driven approaches have greatly broadened EC analysis Chen & Zhang (2023); Dai et al. (2025). Graph neural networks (GNNs) exploit topological information to infer causal dependencies, recurrent neural networks (RNNs) capture temporal dynamics and nonlinear lags, and generative architectures such as GANs and diffusion models have been studied for augmentation and causal structure discovery Liang et al. (2024); Ji et al. (2025); Chen et al. (2025). Although these methods have achieved encouraging results, most efforts remain restricted to unimodal data—typically Functional magnetic resonance imaging (fMRI) and thus fail to leverage the complementary strengths of multimodal integration.

The complementary nature of fMRI and electroencephalography (EEG) presents unique opportunities for enhanced EC learning Wei et al. (2025). fMRI provides high spatial resolution but suffers from low temporal resolution, whereas EEG offers millisecond-level temporal precision Herwig et al. (2003) but relatively poor spatial localization Zhang et al. (2024a); Sakkalis (2011). By in-

tegrating these two modalities, it becomes possible to exploit both fine-grained spatial detail and precise temporal dynamics, which is essential for accurately uncovering the brain's causal organization. However, most existing multimodal approaches fail to fully harness these complementary advantages. For instance, Anwar et al. employed Granger causality to learn EC from multimodal data, but their method did not achieve genuine integration of fMRI and EEG information Muthuraman (2016). Tu et al. proposed a linear state-space model to combine temporal information across modalities, yet this approach is limited in capturing nonlinear causal dependencies Tu et al. (2019). Liu et al. introduced a cross-modal mapping strategy, but it essentially relied on feature concatenation without leveraging spatial or temporal constraints. More broadly, many multimodal methods simply apply concatenation or linear mapping Liu et al. (2024), overlooking anatomical correspondence between EEG electrodes and brain regions, as well as the temporal dynamics of neural activity. Such limitations reduce their interpretability and hinder generalization, highlighting the need for methods that explicitly enforce structural consistency and model nonlinear cross-modal interactions.

To overcome these challenges, we propose STBO-EC, a spatially and temporally guided framework for multimodal EC learning based on Bayesian optimization. Our framework introduces three innovations: an anatomy-informed spatial alignment strategy that utilizes known brain region coordinates to align EEG electrodes with corresponding fMRI regions, ensuring structural consistency across modalities; a time-slice-based alignment and fusion module that bridges the temporal resolution gap between EEG and fMRI, enabling synchronized and complementary spatiotemporal integration; and the use of a Bayesian optimization method (DrBO) with low-rank parameterization, which efficiently searches the exponential EC space and provides uncertainty-aware causal inference.

Our contributions are summarized as follows:

- We propose a neuroanatomy-informed spatial alignment strategy that enforces structural consistency between EEG and fMRI at the brain-region level.
- We design a time-slice-based alignment and fusion module that synchronizes multimodal signals, enabling effective integration of complementary temporal and spatial information.
- We adopt a Bayesian optimization-based causal structure learning method (DrBO) with low-rank parameterization, enabling efficient modeling of nonlinear EC.
- We validate our framework on two real EEG–fMRI datasets, and results show that STBO-EC consistently outperforms state-of-the-art baselines across multiple evaluation metrics.

## 2 PRELIMINARY AND RELATED WORK

### 2.1 BRAIN EFFECTIVE CONNECTIVITY

Brain effective connectivity (EC) characterizes the directed causal influence exerted by one brain region on another, thereby reflecting both the directionality and strength of information flow in the brain Cai et al. (2018); Huang et al. (2018). Formally, EC can be represented as a directed graph $G = (V, E, W)$, where $V$ denotes the set of brain regions (ROIs), $E$ is the set of directed edges corresponding to causal interactions, and $W \in \mathbb{R}^{|V| \times |V|}$ is an asymmetric weighted adjacency matrix. Each element $W_{ij}$ quantifies the causal effect from region $i$ to region $j$, thereby encoding both the topology and strength of directional connectivity within the network.

### 2.2 MULTIMODAL EFFECTIVE CONNECTIVITY

Multimodal EC extends traditional EC by integrating complementary information from multiple neuroimaging modalities (e.g., EEG and fMRI) to enhance causal inference in brain networks dau (2011). This concept emerged from early EEG-fMRI fusion studies Debener et al. (2006); Sato et al. (2010), where joint modeling of different modalities improved the stability of causal estimation. Recent advances have formalized multimodal EC as learning cross-modal consistent patterns of directed brain interactions from multi-source neuroimaging data Liu et al. (2024); Li et al. (2025), providing a more comprehensive framework for studying complex cognitive processes and psychiatric disorders.

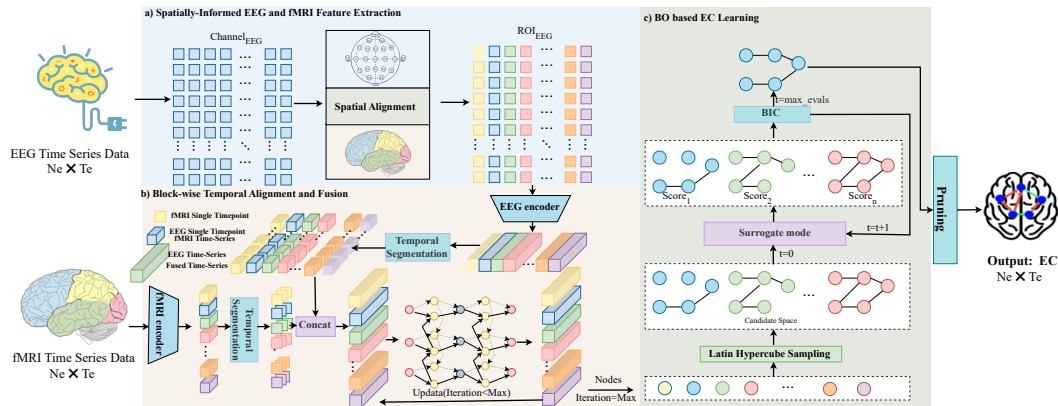

Figure 1: Workflow of the proposed STBO-EC framework, which consists of three main modules: (a) spatially-informed EEG and fMRI feature extraction, (b) block-wise temporal alignment and fusion, and (c) BO-based EC learning.

## 2.3 DAG RECOVERY VIA BAYESIAN OPTIMIZATION

DrBO (DAG recovery via Bayesian Optimization) is a causal structure learning method that applies Bayesian Optimization (BO) to recover directed acyclic graphs (DAGs) from purely observational data Duong et al.. It reformulates the original combinatorial optimization problem into a low-dimensional continuous one by representing a DAG with a node potential vector and a low-rank edge embedding matrix. This reduces the search space from $\mathcal{O}(d^2)$ to $\mathcal{O}(d(1 + k))$, where $d$ is the number of nodes and $k$ is the embedding dimension. To model the acquisition function in BO, DrBO employs dropout neural networks as surrogate models, enabling uncertainty-aware score estimation via Thompson sampling. Instead of predicting the global DAG score directly, DrBO models node-wise local score components and then aggregates them according to a decomposable scoring function. This approach allows the generation of valid DAGs without explicit acyclicity constraints and guides the search toward high-scoring structures within the BO framework.

## 3 METHOD

We propose STBO-EC, a spatially and temporally guided Bayesian optimization for multi-modal EC learning (as showed in Fig. 1). Our approach first aligns EEG and fMRI data at the brain region level using neuroanatomical priors and spatial mapping, ensuring structural consistency across modalities. We then introduce a time-slice-based alignment and fusion module to comprehensively integrate complementary temporal and spatial information from both EEG and fMRI. Finally, we employ DrBO, a Bayesian optimization-based causal structure learning method with low-rank EC parameterization and a dropout neural network surrogate, to efficiently and interpretably model non-linear causal relationships between brain regions.

## 3.1 NEUROANATOMY-GUIDED SPATIAL ALIGNMENT

The fundamental challenge in EEG-fMRI integration lies in bridging the spatial gap between scalp electrodes and brain regions. While EEG captures electrical activity at $N_e$ scalp locations, fMRI measures hemodynamic responses within $N_f$ anatomically-defined brain regions. We address this challenge through a biophysically-motivated projection that respects the underlying neuroanatomy. Let the EEG data be represented as

$$X_{\text{EEG}} \in \mathbb{R}^{N_e \times T_e}, \tag{1}$$

where $N_e$ is the number of electrodes and $T_e$ is the number of samples. Similarly, the fMRI data is

$$X_{\text{fMRI}} \in \mathbb{R}^{N_f \times T_f}, \tag{2}$$

where $N_f$ is the number of ROIs and $T_f$ is the number of fMRI time frames. Our goal is to project EEG signals into the ROI space of fMRI.

Each electrode $c$ and ROI $i$ has anatomical coordinates $\mathbf{p}_c, \mathbf{q}_i \in \mathbb{R}^3$. Their Euclidean distance is defined as

$$d_{ic} = \|\mathbf{q}_i - \mathbf{p}_c\|_2. \tag{3}$$

We then define a Gaussian kernel weighting:

$$w_{ic} = \frac{\exp(-d_{ic}^2/2\sigma^2)}{\sum_{c'=1}^{N_e} \exp(-d_{ic'}^2/2\sigma^2)}, \tag{4}$$

where $\sigma > 0$ is the diffusion parameter. Normalization ensures $\sum_{c=1}^{N_e} w_{ic} = 1$.

The EEG signal at ROI $i$ and time $t$ is projected as

$$Y_{\text{EEG}}(i,t) = \sum_{c=1}^{N_e} w_{ic} X_{\text{EEG}}(c,t). \tag{5}$$

After spatial alignment, we obtain the ROI-level EEG representation: $Y_{\text{EEG}} \in \mathbb{R}^{N_f \times T_e}$ which is spatially aligned with $X_{\text{fMRI}}$.

## 3.2 BLOCK-WISE TEMPORAL ALIGNMENT AND FUSION

EEG and fMRI differ significantly in temporal resolution, with $T_e \gg T_f$. We address this mismatch using block-wise alignment.

We divide $Y_{\text{EEG}}$ into $T_f$ non-overlapping blocks, each of length

$$L = \frac{T_e}{T_f}. \tag{6}$$

The $j$-th EEG block is

$$B_{\text{EEG}}^{(j)} = Y_{\text{EEG}}(:, (j-1)L + 1 : jL)^\top \in \mathbb{R}^{L \times N_f}. \tag{7}$$

The $j$-th fMRI vector $x_{\text{fMRI}}^{(j)} \in \mathbb{R}^{N_f}$ is expanded to match block length as

$$B_{\text{fMRI}}^{(j)} = \mathbf{1}_L \cdot (x_{\text{fMRI}}^{(j)})^\top \in \mathbb{R}^{L \times N_f}, \tag{8}$$

where $\mathbf{1}_L$ is a column vector of ones.

We then construct the fusion target as

$$B_{\text{target}}^{(j)} = \alpha B_{\text{EEG}}^{(j)} + (1-\alpha) B_{\text{fMRI}}^{(j)}, \quad \alpha \in [0,1]. \tag{9}$$

In our experiments, we fix $\alpha = 0.5$ to enforce equal contribution of both modalities.

A nonlinear mapping $f_\theta$ is trained to approximate

$$\hat{B}_{\text{fused}}^{(j)} = f_\theta(B_{\text{EEG}}^{(j)}, B_{\text{fMRI}}^{(j)}), \tag{10}$$

where $\theta$ are trainable parameters.

The objective is optimized iteratively using gradient descent:

$$\mathcal{L}_{\text{total}}(\theta) = \frac{1}{T_f L N_f} \sum_{j=1}^{T_f} \sum_{t=1}^{L} \sum_{i=1}^{N_f} \left( \hat{b}_{\text{fused}}^{(j)}(t,i) - B_{\text{target}}^{(j)}(t,i) \right)^2, \tag{11}$$

$$\theta^{(k+1)} = \theta^{(k)} - \eta \nabla_\theta \mathcal{L}_{\text{total}}(\theta^{(k)}), \tag{12}$$

where $\eta$ is the learning rate and $k$ is the iteration index.

Finally, sliding-window inference is applied. Let $L = \text{TR} \times \text{Hz}$ denote the EEG samples per fMRI TR, and $\mathcal{W}_t$ the set of windows covering time $t$. The fused output is

$$y_{\text{fused}}(t,i) = \frac{1}{|\mathcal{W}_t|} \sum_{w \in \mathcal{W}_t} \hat{b}_{\text{fused}}^{(w)}(t - w_{\text{start}} + 1, i). \tag{13}$$

The final multimodal sequence is

$$Y_{\text{fused}} \in \mathbb{R}^{N_f \times T_e}, \tag{14}$$

which inherits both the high temporal resolution of EEG and the stable baselines of fMRI.

## 3.3 BO-BASED EC LEARNING

Given the fused multimodal representation $Y_{\text{fused}}$, the objective is to infer the directed EC network among $N_f$ brain regions.

We adopt a Bayesian information criterion (BIC)-based score to evaluate candidate structures:

$$S(D, G) = \ln p(D|\hat{\theta}, G) - \frac{|E|}{2} \ln n, \qquad (15)$$

where $D = Y_{\text{fused}}$, $\hat{\theta}$ are maximum likelihood parameters, $|E|$ is the number of edges, and $n$ is the sample size. This criterion balances model fit and complexity.

To reduce search space, the adjacency matrix is parameterized in a continuous low-rank form:

$$\tilde{A}_{ij} = \sigma(\langle r_i, r_j \rangle), \qquad (16)$$

where $r_i \in \mathbb{R}^k$ is the latent embedding of region $i$, and $\sigma(\cdot)$ is the sigmoid function. A topological ordering vector $p$ enforces acyclicity, leading to

$$A = \tau(p, R). \qquad (17)$$

We cast EC learning as

$$z^* = \arg\max_z S(D, \tau(z)), \qquad (18)$$

where $z = (p, R)$ are continuous DAG parameters. Since $S(\cdot)$ is expensive to compute, we employ Bayesian Optimization with a dropout neural network surrogate $g_\phi(z)$, which estimates both score $\mu_\phi(z)$ and uncertainty $\sigma_\phi(z)$. The acquisition function is

$$a(z) = \mu_\phi(z) + \beta \sigma_\phi(z), \qquad (19)$$

where $\beta$ controls exploration.

By iteratively maximizing $a(z)$ and updating $z$, we obtain the optimal adjacency matrix $\hat{A}$. The inferred EC network is

$$\hat{G} = (V, \hat{E}, \hat{W}), \qquad (20)$$

where $\hat{E}$ denotes the directed edges and $\hat{W}$ their estimated weights.

## 3.4 ALGORITHM DESCRIPTION

The proposed STBO-EC framework integrates EEG and fMRI data through three sequential modules: (1) **Neuroanatomy-Guided Spatial Alignment**, which projects EEG signals into the fMRI ROI space using Gaussian-kernel weighting based on anatomical distances; (2) **Block-wise Temporal Alignment and Fusion**, which synchronizes the temporal scales of EEG and fMRI via a time-slice fusion mechanism, generating multimodal representations that combine EEG's high temporal resolution with fMRI's spatial stability; and (3) **BO-based EC learning**, which performs low-rank parameterized causal discovery using dropout neural network surrogates for scalable and uncertainty-aware inference. The overall pseudocode of STBO-EC is summarized in Algorithm 1.

## 4 EXPERIMENTS

To evaluate the effectiveness of STBO-EC, we apply our proposed method to two publicly available real-world datasets with simultaneously recorded fMRI and EEG data. These experiments are designed to demonstrate the practical applicability and performance advantages of STBO-EC in real multimodal neuroimaging scenarios.Since ground truth effective connectivity (EC) is not available for real datasets, we further assess the quality of EC learning by using the inferred connectivity features to classify brain activity according to known experimental labels. A higher classification accuracy indicates more accurate EC learning.For fair comparison, all methods were evaluated using the same classification pipeline: a 5-fold cross-validated random forest classifier for the Visual Categorization Dataset (6 regions), and a 10-fold cross-validated KNN classifier for the XP2 Dataset (90 regions). These classifiers were chosen to match the dimensionality and sample size of each dataset, ensuring robust and reliable evaluation.

### 4.1 BASELINE METHODS

To comprehensively evaluate the effectiveness of our proposed STBO-EC framework, we compare it with a range of classical and state-of-the-art EC learning methods. The baselines include Patel Patel et al. (2006), pwLiNGAM Hyvarinen (2010), lsGC DSouza et al. (2017), DiffAN Sanchez et al., MetaRLEC Zhang et al. (2024b), MCAN Liu et al. (2024), FSTA-EC Xiong et al. (2025), and our proposed STBO-EC. Among these, MCAN is also a multimodal method that integrates both EEG and fMRI data for EC learning. For all comparison methods, we follow the parameter settings recommended in the original literature and further fine-tune the hyperparameters on a subset of subjects to ensure optimal performance. The evaluation is conducted on the Visual Categorization Dataset, and we report a comprehensive set of graph and classification metrics, including Accuracy, Precision, Recall, F1 score, Specificity, Error Rate, and ROC_AUC POWERS (2011); Fawcett (2006). The detailed description of the baseline methods and evaluation metrics can be found in Appendix A.1 and Appendix A.2. The detailed results are summarized in Table 1 and Table 2.

### 4.2 DATA DESCRIPTION

We evaluate our method on two public multimodal neuroimaging datasets: the visual categorization dataset and the XP2 motor imagery neurofeedback dataset. Details of the datasets are provided in Appendix A.3, and the data preprocessing procedure is described in Appendix A.4.

**Visual categorization dataset**[1]. This dataset includes simultaneous fMRI-EEG recordings from 31 subjects performing an event-related three-class visual categorization task (face, car, house) Tu et al. (2019). Each subject completed four runs, with 180 trials per run (60 per category), resulting in a total of 93 multimodal samples. The fMRI data consist of time series extracted from 6 task-related ROIs (FFA, PPA, SPL, ACC, PMC, and bilateral FEF), with a TR of 2 seconds, while EEG was recorded from 34 MR-compatible channels and subsequently downsampled to 10 Hz. This dataset provides a benchmark for evaluating EC models under well-controlled experimental conditions with simultaneous high spatial and temporal resolution signals.

**XP2 dataset**[2]. The XP2 dataset Lioi et al. (2019) contains simultaneous fMRI-EEG data from 40 decoupled Neurofeedback (dNF) and 28 motor imagery (MI) participants. During the task, subjects alternated between 20-second blocks of rest and task across five runs. EEG was recorded from 64 channels at 5 kHz (downsampled to 200 Hz after preprocessing in EEGLAB), with FCz as the reference and AFz as the ground. Simultaneous fMRI was acquired on a 3T Siemens Verio scanner (EPI sequence, TR = 1s, TE = 23 ms, resolution = $3 \times 3 \times 3$ mm$^3$), covering the superior half of the brain. Regional time series were extracted from AAL-defined ROIs following standard preprocessing with DPABI. This dataset offers a challenging testbed for multimodal EC analysis due to its motor imagery paradigm, neurofeedback design, and high-density EEG-fMRI recordings.

### 4.3 MODEL CONFIGURATION

The main hyperparameters of the STBO-EC model are as follows. We set the batch size $B$ to 64 and the EC rank $k$ to 8. The number of training steps ($n_{\text{grads}}$) is set to 10, and the number of preliminary candidates ($C$) is 100,000. The maximum number of iterations is set to 2,000. The hyperparameter details are provided in Appendix A.6.

### 4.4 ABLATION STUDY

To further evaluate the impact of multimodal integration in STBO-EC, we conducted ablation experiments using only the fMRI modality as input, while keeping all other settings unchanged. As shown in Tables 1 and 2, on both datasets, the multimodal version of STBO-EC consistently outperforms its fMRI-only counterpart, denoted as DrBO, across all key metrics. The performance gain is particularly pronounced on the XP2 dataset, which contains more brain regions and richer multimodal signals. These results demonstrate the effectiveness of integrating EEG and fMRI data for more accurate effective connectivity learning and brain state classification.

---

[1]https://github.com/taotu/VBLDS_Connectivity_EEG_fMRI
[2]https://openneuro.org/datasets/ds002338/versions/2.0.2

| Methods (Years) | Accuracy↑ | Precision↑ | Recall↑ | F1↑ | Error_Rate↓ | Specificity↑ | ROC_AUC↑ |
|---|---|---|---|---|---|---|---|
| Patel (2006) | $0.27 \pm 0.13$ | $0.25 \pm 0.13$ | $0.27 \pm 0.14$ | $0.26 \pm 0.13$ | $0.73 \pm 0.13$ | $0.27 \pm 0.14$ | $0.50 \pm 0.11$ |
| lsGC (2010) | $\underline{0.39} \pm 0.11$ | $\underline{0.41} \pm 0.13$ | $\underline{0.39} \pm 0.11$ | $\underline{0.38} \pm 0.11$ | $\underline{0.61} \pm 0.11$ | $\underline{0.39} \pm 0.11$ | $\underline{0.56} \pm 0.06$ |
| pwLiNGAM (2017) | $0.31 \pm 0.09$ | $0.31 \pm 0.10$ | $0.31 \pm 0.09$ | $0.30 \pm 0.08$ | $0.69 \pm 0.09$ | $0.31 \pm 0.09$ | $0.54 \pm 0.08$ |
| DiffAN (2023) | $0.33 \pm 0.10$ | $0.35 \pm 0.11$ | $0.33 \pm 0.10$ | $0.33 \pm 0.10$ | $0.67 \pm 0.10$ | $0.33 \pm 0.10$ | $0.56 \pm 0.08$ |
| MetaRLEC (2024) | $0.35 \pm 0.16$ | $0.37 \pm 0.20$ | $0.35 \pm 0.16$ | $0.34 \pm 0.15$ | $0.65 \pm 0.16$ | $0.35 \pm 0.16$ | $0.47 \pm 0.10$ |
| MCAN (2024) | $0.27 \pm 0.06$ | $0.24 \pm 0.08$ | $0.27 \pm 0.07$ | $0.25 \pm 0.07$ | $0.73 \pm 0.06$ | $0.27 \pm 0.07$ | $0.49 \pm 0.09$ |
| FSTA-EC (2025) | $0.26 \pm 0.11$ | $0.23 \pm 0.11$ | $0.26 \pm 0.11$ | $0.24 \pm 0.11$ | $0.74 \pm 0.11$ | $0.26 \pm 0.11$ | $0.41 \pm 0.08$ |
| DrBO (2025) | $0.38 \pm 0.10$ | $0.40 \pm 0.10$ | $0.38 \pm 0.10$ | $0.37 \pm 0.10$ | $0.62 \pm 0.10$ | $0.38 \pm 0.10$ | $0.49 \pm 0.14$ |
| **STBO-EC (Ours)** | $\mathbf{0.42} \pm 0.11$ | $\mathbf{0.49} \pm 0.14$ | $\mathbf{0.42} \pm 0.11$ | $\mathbf{0.42} \pm 0.11$ | $\mathbf{0.58} \pm 0.11$ | $\mathbf{0.42} \pm 0.11$ | $\mathbf{0.57} \pm 0.10$ |

Table 1: Performance comparison of different methods across various metrics on the Visual Categorization Dataset. The best results are in **bold**, and the second-best are underlined. For Error_Rate, the lowest and second lowest are highlighted.

## 5 EXPERIMENTAL RESULTS

### 5.1 RESULTS ON VISUAL CATEGORIZATION DATASET

The results on the Visual Categorization Dataset (Table 1) clearly demonstrate the superior performance of STBO-EC across all key evaluation metrics. Specifically, STBO-EC achieves the highest accuracy (0.42), precision (0.49), recall (0.42), and F1 score (0.42) among all compared methods. Notably, the second-best method, lsGC, attains an accuracy of 0.39 and an F1 score of 0.38, which are both lower than those of STBO-EC. While some methods, such as pwLiNGAM and DiffAN, show moderate performance in certain metrics, they fall short in terms of overall balanced performance. For example, pwLiNGAM achieves a relatively high precision (0.31) but lower recall (0.31) and F1 (0.30), indicating a tendency to miss true connections. Similarly, DiffAN yields balanced but generally lower scores across all metrics. The $p$-values for these comparisons are also provided in Appendix A.5.

It is also worth noting that the overall classification accuracies for all methods are not very high on this dataset. This is primarily due to the intrinsic similarity between the Car and House conditions, which makes it inherently challenging to distinguish between these two categories based on effective connectivity patterns. As a result, even advanced multimodal methods like MCAN do not show a clear advantage over single-modal approaches in this scenario.

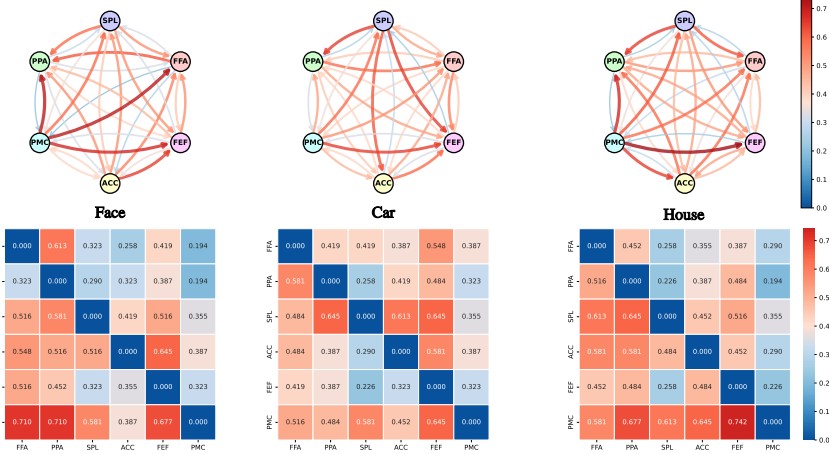

Figure 2: Estimated EC networks for six brain regions on the Visual Categorization Dataset. The node positions in the figure are for illustration only and do not precisely reflect real anatomical locations.

The physiological analysis of the EC (as showed in Figure 2) further supports these findings. The connectivity patterns and strengths differ across tasks. For example, in the Face task, the FFA exhibits strong connections with motor and attention-related regions (such as PMC and FEF), suggesting that face processing involves motor preparation and attentional modulation. In contrast, in the Car and House tasks, the PPA shows more prominent connections with other regions, reflecting the importance of scene and spatial information in these tasks. Overall, these results reveal the brain's flexible regulation of information flow and functional coordination during different visual cognitive processes.

## 5.2 RESULTS ON XP2 DATASET

On the XP2 motor imagery neurofeedback dataset (Table 2), STBO-EC again demonstrates clear superiority across multiple evaluation metrics. It achieves the highest accuracy (0.62), precision (0.53), and ROC_AUC (0.64), and ranks second in recall (0.62) and F1 score (0.55) (as showed in Figure 6). Compared to the strongest alternative methods, such as FSTA-EC and pwLiNGAM, STBO-EC delivers significantly higher accuracy and precision, as confirmed by paired significance tests. The $p$-values for these comparisons are also provided in Appendix A.5. Furthermore, STBO-EC consistently outperforms the single-modal baselines (Patel, lsGC, and DiffAN) on at least three core metrics, highlighting the advantage of joint spatio-temporal modeling of EEG and fMRI data.

These results demonstrate that STBO-EC achieves a better balance between sensitivity and specificity, leading to more robust and reliable classification performance. The consistent statistical significance across multiple metrics further underscores the effectiveness of the proposed multimodal approach. Note that MCAN, as a multimodal method, was not included on the XP2 dataset due to its high computational cost on large-scale brain networks.

| Methods (Years) | Accuracy↑ | Precision↑ | Recall↑ | F1↑ | Error_Rate↓ | Specificity↑ | ROC_AUC↑ |
|---|---|---|---|---|---|---|---|
| Patel (2006) | $0.43 \pm 0.17$ | $0.39 \pm 0.15$ | **0.70** $\pm 0.23$ | $0.49 \pm 0.16$ | $\underline{0.57} \pm 0.17$ | $0.25 \pm 0.24$ | $0.47 \pm 0.20$ |
| lsGC (2010) | $\underline{0.61} \pm 0.06$ | $0.10 \pm 0.32$ | $0.03 \pm 0.11$ | $0.05 \pm 0.16$ | $0.39 \pm 0.06$ | **1.00** $\pm 0.00$ | $0.53 \pm 0.22$ |
| pwLiNGAM (2017) | $0.61 \pm 0.18$ | $\underline{0.47} \pm 0.41$ | $0.32 \pm 0.30$ | $0.35 \pm 0.31$ | $0.39 \pm 0.18$ | $\underline{0.80} \pm 0.23$ | $\underline{0.57} \pm 0.24$ |
| DiffAN (2023) | $0.54 \pm 0.15$ | $0.30 \pm 0.36$ | $0.22 \pm 0.25$ | $0.25 \pm 0.28$ | $0.46 \pm 0.15$ | $0.75 \pm 0.17$ | $0.54 \pm 0.18$ |
| MetaRLEC (2024) | $0.52 \pm 0.18$ | $0.37 \pm 0.40$ | $0.30 \pm 0.33$ | $0.29 \pm 0.28$ | $0.48 \pm 0.18$ | $0.65 \pm 0.27$ | $0.50 \pm 0.21$ |
| MCAN (2024) | – | – | – | – | – | – | – |
| FSTA-EC (2025) | $0.54 \pm 0.25$ | $0.40 \pm 0.47$ | $0.25 \pm 0.29$ | $0.30 \pm 0.34$ | $0.46 \pm 0.25$ | $0.73 \pm 0.30$ | $0.50 \pm 0.32$ |
| DrBO (2025) | $0.53 \pm 0.14$ | $0.35 \pm 0.22$ | $0.43 \pm 0.26$ | $0.38 \pm 0.23$ | $0.47 \pm 0.14$ | $0.58 \pm 0.24$ | $0.49 \pm 0.11$ |
| **STBO-EC (Ours)** | **0.62** $\pm 0.15$ | **0.53** $\pm 0.26$ | $\underline{0.62} \pm 0.29$ | **0.55** $\pm 0.23$ | **0.38** $\pm 0.15$ | $0.63 \pm 0.18$ | **0.64** $\pm 0.13$ |

Table 2: Performance comparison of different methods across various metrics on the XP2 Dataset. The best results are in **bold**, and the second-best are underlined (for Error_Rate, the lowest and second lowest are highlighted).

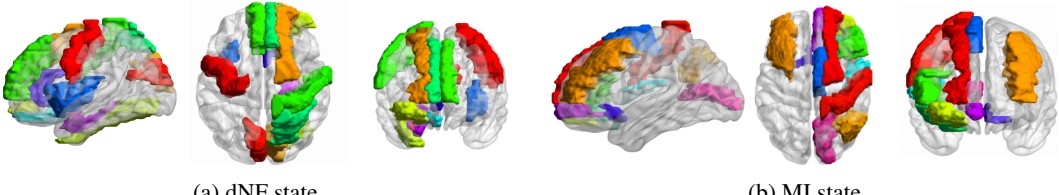

(a) dNF state          (b) MI state

Figure 3: Top 10 most active brain regions in the dNF and MI states. The highlighted nodes correspond to the most active regions identified in each condition.

In the dNF state, the motor regions (such as Rolandic_Oper_R, Supp_Motor_Area_R, and Postcentral_R), as well as certain frontal and parietal areas (including Frontal_Mid_R, Angular_R, and Supra-Marginal_R), and parts of the occipital and temporal lobes (such as Calcarine_R, Fusiform_R, and Heschl_R), exhibit relatively high out-degree and in-degree values. This indicates that, even in the resting or baseline state, the brain maintains active information flow and coordination within networks related to motor, sensory, visual, and auditory functions. Additionally, regions within the lim-

bic system and basal ganglia (such as Insula_R, ParaHippocampal_R, Putamen_R, and Thalamus_R) also show strong connectivity, suggesting that emotion, memory, and motor regulation are functionally engaged. Frontal lobe regions (e.g., Frontal_Sup_R, Frontal_Mid_Orb_R) show substantial connectivity, indicating executive and attentional processes even without explicit motor tasks. Parietal (Angular_R, SupraMarginal_R) and occipital (Calcarine_R, Fusiform_R) regions are also highly connected, supporting sensory integration and visual processing. Additionally, temporal (Heschl_R, Temporal_Sup_R), limbic, and basal ganglia areas (Insula_R, Putamen_R, Thalamus_R) are engaged, reflecting involvement in perception, emotion, and motor control during baseline.

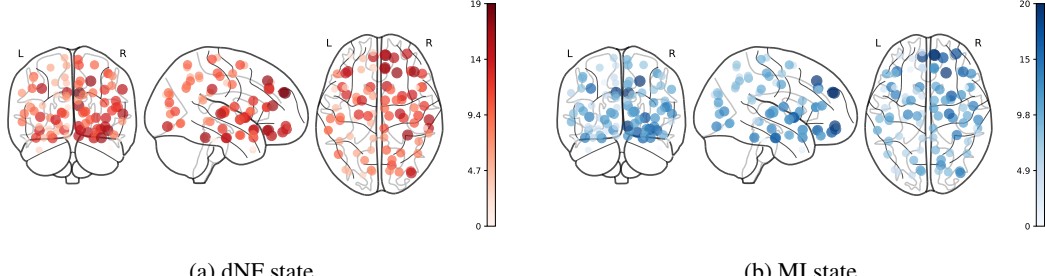

(a) dNF state          (b) MI state

Figure 4: Out-degree distribution of brain regions in (a) dNF state and (b) MI state. Node size and color represent the out-degree values in the EC network.

In the MI state, the out-degree and in-degree values of the motor, frontal, parietal, and occipital regions are further increased, especially in areas such as Rolandic_Oper_R, Frontal_Sup_R, Frontal_Mid_Orb_R, and Fusiform_R. This reflects the significant activation of neural networks associated with motor preparation, execution, spatial perception, and visual processing during the motor imagery task. At the same time, certain regions within the limbic system and basal ganglia (such as ParaHippocampal_R, Putamen_R, and Pallidum_L) also demonstrate enhanced connectivity, indicating a greater involvement of emotion, memory, and motor regulation during motor imagery. The occipital lobe (e.g., Calcarine_R, Fusiform_R) shows increased connectivity, likely linked to visual imagery and integration with motor representations. Temporal regions (Heschl_R, Temporal_Sup_R) remain active, supporting auditory and semantic processing. Enhanced connectivity in limbic and basal ganglia areas (ParaHippocampal_R, Putamen_R, Pallidum_L) suggests greater involvement of memory, emotion, and motor control during motor imagery.

Overall, statistical analysis of the EC matrices reveals that the MI state is characterized by stronger coordination and information flow across multiple functional brain networks. The integration of motor, cognitive, sensory, and emotional systems becomes more pronounced, highlighting the extensive recruitment and synchronization of brain regions during motor imagery. These findings provide important physiological evidence for understanding the neural mechanisms underlying motor-related cognition and regulation. More detailed descriptions are provided in Appendix A.7.

## 6   CONCLUSION AND LIMITATIONS

In summary, we presented STBO-EC, a framework for multimodal EC that integrates neuroanatomical spatial alignment, temporal synchronization, and Bayesian optimization for causal discovery. By leveraging the complementary strengths of EEG and fMRI, our method achieves consistent improvements over state-of-the-art baselines across multiple datasets. These results highlight that respecting the distinct characteristics of each modality, rather than applying naive fusion, is crucial for uncovering the brain's causal organization. Moreover, the anatomically grounded and uncertainty-aware design of STBO-EC provides new opportunities for understanding neural mechanisms in both healthy cognition and neurological disorders.

Despite these advances, our method still faces some limitations. In detail, the accuracy of anatomical mapping and temporal alignment may be affected by individual variability, and the scalability to larger or higher-resolution datasets requires further exploration. In the future work, we plan to enhance the robustness and generalizability of STBO-EC, and to validate its effectiveness in broader cognitive and clinical applications.

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

# A APPENDIX

## A.1 BASELINE METHODS AND OPERATING ENVIRONMENT

To thoroughly assess the effectiveness of the proposed STBO-EC framework, we compare it against a suite of classical and state-of-the-art EC learning methods. The baselines include Patel, pwLiNGAM, lsGC, DiffAN, MetaRLEC, MCAN, FSTA-EC, and our proposed STBO-EC. Notably, MCAN is a multimodal approach that integrates both EEG and fMRI data for EC learning. For all baseline methods, we adopt the parameter settings recommended in the original publications and further fine-tune hyperparameters on a subset of subjects to ensure fair and optimal performance.The full list of baseline methods and their parameter settings is summarized in Table 3, which provides the parameter settings for all compared methods.

All experiments are conducted in a robust computational environment, comprising both advanced hardware and a comprehensive software stack.The hardware setup consists of a high-performance NVIDIA RTX 5090 GPU with 32 GB of memory, an NVIDIA GeForce RTX 3090 GPU, and an AMD Ryzen 9 5950X 16-core CPU. These components collectively provide a powerful and efficient computing platform. The system operates on Ubuntu 20.04.6 LTS, ensuring a stable and robust foundation for all experimental procedures. Additionally, the machine is equipped with 128 GB of high-speed DDR4 RAM, which guarantees excellent responsiveness and computational throughput.

The software environment is built around Python 3.10, with key scientific and deep learning libraries including NumPy 1.26.4, pandas 2.2.3, scikit-learn 1.6.1, matplotlib 3.10.3, seaborn 0.13.2, PyTorch 2.7.1+cu128, JAX 0.6.2, and jaxlib 0.6.2. All deep learning computations are accelerated using CUDA version 12.8, ensuring optimal utilization of GPU resources. This comprehensive combination of hardware and software provides a reliable and reproducible environment for our research.

| Method | Parameter settings |
|---|---|
| Patel | threshold = 0.3 |
| lsGC | cmp = 5, ARorder = 2, normalize = 1 |
| pwLiNGAM | method = 1 |
| DiffAN | lr = 0.001, $\beta_{start}$ = 0.0001, $\beta_{end}$ = 0.02, nh = 1024 |
| MetaCAE | nh=64, $\alpha$=0.05, $\beta$=20.0, $k$=3, $d$=4, $lr_1$=0.02, $lr_2$=0.02, $lr_3$=0.001, $lr_{main}$=0.002 |
| MCAN | $\alpha$=0.4, $\beta$=0.2, $\gamma$=0.4 |
| FSTA-EC | Adam ($\beta_1$=0.90, $\beta_2$=0.98), embed=16, dropout=0.2, attention heads=2 |

Table 3: Parameter settings for all compared methods.

## A.2 EVALUATION METRICS

Since the datasets used in this study do not provide ground-truth EC networks, we evaluate the quality of the learned EC by using it as input features for downstream classification tasks. Specifically, we assess the classification performance using the following standard metrics: Accuracy, Precision, Recall, F1 score, Specificity, and ROC_AUC.

Let $TP$ (true positives), $TN$ (true negatives), $FP$ (false positives), and $FN$ (false negatives) denote the elements of the confusion matrix. The evaluation metrics are defined as follows:

$$\text{Accuracy} = \frac{TP + TN}{TP + TN + FP + FN}, \tag{21}$$

$$\text{Precision} = \frac{TP}{TP + FP}, \tag{22}$$

$$\text{Recall} = \frac{TP}{TP + FN}, \tag{23}$$

$$\text{Specificity} = \frac{TN}{TN + FP}, \tag{24}$$

$$\text{F1 score} = \frac{2 \cdot \text{Precision} \cdot \text{Recall}}{\text{Precision} + \text{Recall}}, \tag{25}$$

$$\text{ROC\_AUC} = \int_0^1 \text{TPR}(t) \, d\text{FPR}(t), \tag{26}$$

where $\text{TPR}(t)$ and $\text{FPR}(t)$ denote the true positive rate and false positive rate at threshold $t$, respectively.

**Accuracy** measures the proportion of correctly classified samples among all samples. **Precision** is the proportion of true positive predictions among all positive predictions, reflecting the model's ability to avoid false positives. **Recall** is the proportion of true positive samples that are correctly identified, indicating the model's ability to detect actual positives. **Specificity** is the proportion of true negative samples that are correctly identified, reflecting the model's ability to avoid false positives. **F1 score** is the harmonic mean of Precision and Recall, providing a balanced measure of both. **Error Rate** measures the proportion of incorrectly classified samples among all samples, reflecting the model's overall misclassification rate. **ROC\_AUC** (Area Under the Receiver Operating Characteristic Curve) summarizes the trade-off between true positive and false positive rates across different thresholds, providing an overall measure of discriminative ability.

All metrics are reported such that higher values indicate better classification performance ($\uparrow$), except for Error Rate, where lower values indicate better performance ($\downarrow$). These metrics comprehensively evaluate the effectiveness of the learned EC features in supporting downstream classification tasks.

### A.3 DATASET SUMMARY AND AVAILABILITY

**Visual Categorization Dataset.** The visual categorization dataset used in this study was collected while subjects performed an event-related three-choice visual categorization task, with simultaneous EEG and fMRI acquisition. Each subject completed four runs of the task, and each run consisted of 180 trials (60 for each category: face, car, and house), with each image presented for 2 seconds per trial. In total, there are 31 groups of simultaneous fMRI-EEG data for each category, resulting in 93 samples.

The fMRI data cover 6 regions of interest (ROIs): the fusiform face area (FFA), parahippocampal place area (PPA), superior parietal lobule (SPL), anterior cingulate cortex (ACC), premotor cortex (PMC), and bilateral frontal eye field (FEF). These ROIs were identified based on group-level EEG-informed fMRI analysis and transformed into each subject's native anatomical space. EEG was recorded from 34 channels, and the fMRI repetition time (TR) was set to 2 seconds. Both EEG and fMRI data were preprocessed. The fMRI data have dimensions $6 \times 56000$, and the EEG data have dimensions $34 \times 56000$. To reduce computational complexity, the EEG sampling rate was downsampled to 10 Hz. For analysis, the three categorization tasks are concatenated every 10 trials.

**XP2 Dataset.** The XP2 dataset is a multimodal neuroimaging dataset that provides simultaneous EEG and fMRI recordings during motor imagery neurofeedback tasks. Sixteen healthy subjects participated in the experiment, each performing five runs with a block design alternating between 20 seconds of rest and 20 seconds of task. Subjects were randomly assigned to two groups: one group performed bimodal EEG-fMRI neurofeedback with a bi-dimensional feedback display, while the other group performed the same task with a mono-dimensional feedback display. The dNF task includes 40 subjects, and the MI task includes 28 subjects.

EEG data were recorded using a 64-channel MR-compatible system at a sampling rate of 5 kHz, with FCz as the reference and AFz as the ground. The EEG data were preprocessed using EEGLAB, including artifact removal, filtering, and channel interpolation as needed, and then downsampled to 200 Hz. The preprocessed EEG signals are provided in standard BIDS format, and neurofeedback scores are also available for each subject and session.

fMRI data were acquired simultaneously using a 3T Siemens Verio scanner with an EPI sequence (TR = 1 s, TE = 23 ms, resolution = $3 \times 3 \times 3$ mm$^3$), covering the superior half of the brain. The fMRI

---

**Algorithm 1:** STBO-EC Framework

---

**Input:** EEG data $X_{\text{EEG}} \in \mathbb{R}^{N_e \times T_e}$, fMRI data $X_{\text{fMRI}} \in \mathbb{R}^{N_f \times T_f}$.
**Output:** Brain effective connectivity $G = (V, E, W)$.

**Step 1: Spatial Alignment**
**for** *each ROI $i$ and electrode $c$* **do**
    Compute anatomical distance $d_{ic} = \|\mathbf{q}_i - \mathbf{p}_c\|_2$;
    Compute Gaussian weights $w_{ic} = \frac{\exp(-d_{ic}^2/2\sigma^2)}{\sum_{c'} \exp(-d_{ic'}^2/2\sigma^2)}$;

Project EEG to ROI space: $Y_{\text{EEG}}(i, t) = \sum_c w_{ic} X_{\text{EEG}}(c, t)$;

**Step 2: Temporal Alignment and Fusion**
Partition $Y_{\text{EEG}}$ into $T_f$ blocks of length $L = T_e/T_f$;
**for** $j = 1$ *to* $T_f$ **do**
    Construct EEG block $B_{\text{EEG}}^{(j)}$;
    Expand fMRI frame $x_{\text{fMRI}}^{(j)}$ to block $B_{\text{fMRI}}^{(j)}$;
    Construct fusion target $B_{\text{target}}^{(j)} = \alpha B_{\text{EEG}}^{(j)} + (1-\alpha) B_{\text{fMRI}}^{(j)}$;
    Train nonlinear mapping $f_\theta$ to minimize MSE to $B_{\text{target}}^{(j)}$;

Obtain fused sequence $Y_{\text{fused}} \in \mathbb{R}^{N_f \times T_e}$;

**Step 3: Bayesian Optimization for EC Learning**
Initialize low-rank parameterization $\mathbf{z} = (\mathbf{p}, \mathbf{R})$;
Initialize surrogate networks $\{\text{DropoutNN}_i\}$ for each node;
**for** $m = 1$ *to* $M$ **do**
    Sample candidate graphs from trust region around $\mathbf{z}$;
    Predict node scores $\hat{s}_i$ using surrogates;
    Select promising graphs via Thompson sampling;
    Evaluate true score $S(\mathcal{D}, A)$ and update replay buffer;
    Update surrogate networks with replay buffer;
    Adapt trust region size;

Prune spurious edges via conditional independence tests;
**return** Final EC $G$;

---

data were preprocessed using DPABI, including slice-timing correction, realignment, coregistration, spatial smoothing, normalization to MNI space, and extraction of regional time series based on the AAL atlas.

### A.4 DATA PREPROCESSING PROCEDURES

**Visual Categorization Dataset.** This dataset provides preprocessed EEG and fMRI data. No additional preprocessing was performed in this study. For EEG analysis, we selected the standard 32-channel electrode configuration.

**XP2 Dataset.** For the XP2 dataset, both EEG and fMRI data underwent a comprehensive preprocessing pipeline to ensure data quality, temporal alignment, and compatibility for multimodal analysis.

For EEG, the raw signals were first loaded and gradient artifacts were corrected using template subtraction based on fMRI volume triggers. The data were then downsampled to 200 Hz and low-pass filtered at 50 Hz to remove high-frequency noise. Cardiac artifacts were removed using the ECG channel and template subtraction. To ensure temporal alignment with the fMRI data, the first 10 time points (corresponding to TR $\times$ Hz samples) were discarded from each EEG recording. The resulting EEG time series were then segmented according to task events, and only the segments corresponding to the fMRI acquisition period were retained for further analysis.

For fMRI, preprocessing was performed using DPABI with the following steps: removal of the first 10 time points, slice-timing correction (16 slices, slice order [2 4 6 8 10 12 14 16 1 3 5 7 9

11 13 15], reference slice 8), realignment, coregistration, spatial normalization to MNI space, and spatial smoothing with a Gaussian kernel (FWHM = 4 mm). The final voxel size was resampled to $3 \times 3 \times 3$ mm$^3$ to ensure consistency across subjects. Segmentation was performed using the DARTEL algorithm, and nuisance covariates (including Friston 24 head motion parameters and signals from white matter, CSF, and global signal) were regressed out. The fMRI time series were then extracted for each region of interest (ROI) based on the Automated Anatomical Labeling (AAL) atlas, which defines 90 cortical and subcortical brain regions. To ensure data consistency, brain regions not fully covered in the acquisition were excluded from further analysis.

This preprocessing pipeline ensures that both EEG and fMRI data are artifact-free, temporally aligned, and region-matched at the level of 90 AAL-defined ROIs.

| Method | Evaluation Metric (p-value) | | | | | | |
|---|---|---|---|---|---|---|---|
| | Accuracy | Precision | Recall | F1 | Error Rate | Specificity | ROC_AUC |
| Patel (2006) | 0.0002 * | 0.0146 * | 0.0021 * | 0.0015 * | 0.4041 | 0.0021 * | 0.0571 |
| lsGC (2010) | 0.3802 | 0.9002 | 0.3802 | 0.0095 * | 0.5648 | 0.3802 | 0.5786 |
| pwLiNGAM (2017) | 0.0005 * | 0.0251 * | 0.0005 * | 0.0037 * | 0.0888 | 0.0005 * | 0.3660 |
| DiffAN (2023) | 0.0023 * | 0.0216 * | 0.0023 * | 0.0023 * | 0.7374 | 0.0023 * | 0.4528 |
| MetaRLEC (2024) | 0.0075 * | 0.5910 | 0.0075 * | 0.0015 * | 0.9913 | 0.0075 * | 0.3305 |
| MCAN (2024) | − | − | − | − | − | − | − |
| FSTA-EC (2025) | 0.0000 * | 0.0000 * | 0.0000 * | 0.0003 * | 0.5328 | 0.0000 * | 0.0001 * |
| STBO-EC (fMRI) | 0.3275 | 0.9932 | 0.3275 | 0.1117 | 0.5187 | 0.3275 | 0.0195 * |

Table 4: P-values of statistical tests for different methods on the Visual Categorization Dataset. $^*p < 0.05$.

| Method | Evaluation Metric (p-value) | | | | | | |
|---|---|---|---|---|---|---|---|
| | Accuracy | Precision | Recall | F1 | Error Rate | Specificity | ROC_AUC |
| Patel (2006) | 0.0182 * | 0.0242 * | 0.0796 | 0.5811 | 0.1967 | 0.0000 * | 0.4102 |
| lsGC (2010) | 0.3856 | 0.0000 * | 0.0000 * | 0.0000 * | 0.5676 | 0.0000 * | 0.6822 |
| pwLiNGAM (2017) | 0.1260 | 0.0006 * | 0.0008 * | 0.0149 * | 0.8825 | 0.7006 | 0.8134 |
| DiffAN (2023) | 0.3957 | 0.0305 * | 0.0001 * | 0.0736 | 0.4615 | 0.0774 | 0.3949 |
| MetaRLEC (2024) | 0.1496 | 0.0203 * | 0.0000 * | 0.0435 * | 0.8685 | 0.5294 | 0.5471 |
| MCAN (2024) | − | − | − | − | − | − | − |
| FSTA-EC (2025) | 0.3934 | 0.6427 | 0.0016 * | 0.0097 * | 0.9910 | 0.1731 | 0.4035 |
| STBO-EC (fMRI) | 0.9225 | 0.1227 | 0.1785 | 0.0005 * | 0.5521 | 0.3163 | 0.0340 * |

Table 5: P-values of statistical tests for different methods on the XP2 Dataset. $^*p < 0.05$.

## A.5 SIGNIFICANCE TEST ANALYSIS

To rigorously assess the performance differences between STBO-EC and other baseline methods, we conducted paired $t$-tests on the classification metrics obtained from cross-validation. For each evaluation metric (Accuracy, Precision, Recall, and F1), a paired $t$-test was performed between the results of STBO-EC and each competing method across the cross-validation folds. A checkmark in Tables 4 and 5 indicates that the improvement of STBO-EC over the corresponding method is statistically significant ($p < 0.05$). These results demonstrate that the observed performance gains of STBO-EC are statistically robust and not attributable to random variation.

## A.6 MODEL CONFIGURATION

The main hyperparameters of the STBO-EC model are as follows. We set the batch size $B$ to 64 and the EC rank $k$ to 8. The number of training steps ($n_{\text{grads}}$) is set to 10, and the number of preliminary candidates ($C$) is 100,000. The model is optimized using the Adam optimizer with a learning rate of 0.1. The replay buffer size ($n_{\text{replay}}$) is set to 1,024. Each neural network layer contains 64 hidden units, and a dropout rate of 0.1 is applied during training.

To further evaluate the robustness of STBO-EC with respect to hyperparameter choices, we conducted a comprehensive parameter sensitivity analysis. Figure 5 illustrates the effects of key hyperparameters (including $n_{\text{grads}}$, $n_{\text{cands}}$, dag_rank, and dropout rate) on both the F1 score and execu-

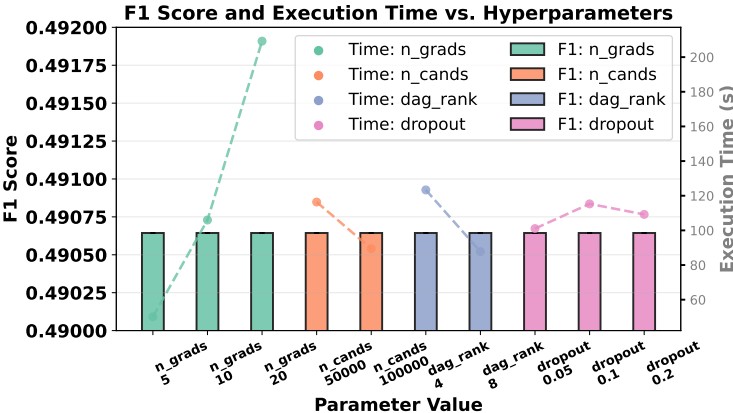

Figure 5: Parameter sensitivity analysis of STBO-EC. Bar plots show the mean F1 score for each hyperparameter value, and line plots indicate the corresponding average execution time.

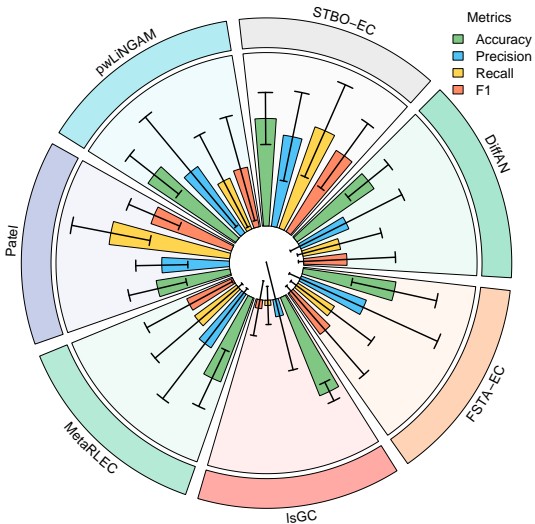

Figure 6: Visualization of Accuracy, Precision, Recall, and F1 for different methods on XP2 Dataset.

tion time. The bar plots show the mean F1 score for each parameter value, while the overlaid line plots indicate the corresponding average execution time. The results demonstrate that the F1 score remains stable across a wide range of parameter settings, highlighting the robustness of STBO-EC to hyperparameter variations. At the same time, execution time increases with certain parameter values, providing practical guidance for parameter selection in real applications.

### A.7 PHYSIOLOGICAL INFORMATION ANALYSIS ON XP2 DATASET

In the dNF state, motor regions such as Rolandic_Oper_R, Supp_Motor_Area_R, and Postcentral_R, as well as frontal (Frontal_Sup_R, Frontal_Mid_Orb_R), parietal (Angular_R, SupraMarginal_R), occipital (Calcarine_R, Fusiform_R), and temporal (Heschl_R, Temporal_Sup_R) areas, exhibit relatively high out-degree and in-degree values. This indicates that, even during rest, the brain maintains active information flow and coordination within networks related to motor, sensory, visual, and auditory functions. Additionally, regions within the limbic system and basal ganglia (such as Insula_R, ParaHippocampal_R, Putamen_R, and Thalamus_R) also show strong connectivity, suggesting ongoing engagement of emotion, memory, and motor regulation.

In the MI state, there is a marked increase in both out-degree and in-degree values across motor, frontal, parietal, and occipital regions, especially in areas such as Rolandic_Oper_R, Frontal_Sup_R,

Frontal_Mid_Orb_R, and Fusiform_R. This reflects the significant activation of neural networks associated with motor preparation, execution, spatial perception, and visual processing during motor imagery. Temporal regions (Heschl_R, Temporal_Sup_R) remain active, supporting auditory and semantic processing. Enhanced connectivity in limbic and basal ganglia areas (ParaHippocampal_R, Putamen_R, Pallidum_L) further suggests greater involvement of memory, emotion, and motor control during motor imagery.

Figure 3 visualizes the top 10 most active brain regions identified in the dNF and MI states. In the dNF state, the most active regions include: Precentral_L, Frontal_Sup_R, Frontal_Mid_Orb_R, Frontal_Sup_Medial_L, Frontal_Sup_Medial_R, Rectus_R, Insula_L, Cingulum_Ant_R, ParaHippocampal_R, Amygdala_R, Cuneus_L, Cuneus_R,Fusiform_R, Postcentral_R, and Parietal_Sup_R. In the MI state, the most active regions are: Frontal_Sup_R, Frontal_Mid_L, Frontal_Mid_Orb_R, Frontal_Inf_Tri_R, Frontal_Inf_Orb_R, Rolandic_Oper_R, Supp_Motor_Area_R, Olfactory_L, Frontal_Med_Orb_R, Calcarine_R, Postcentral_R, and Angular_R. These regions are highlighted on the brain template, illustrating the spatial distribution of the most functionally active areas under each condition.

Overall, statistical analysis of the effective connectivity matrices reveals that the MI state is characterized by stronger coordination and information flow across multiple functional brain networks. The integration of motor, cognitive, sensory, and emotional systems becomes more pronounced, highlighting the extensive recruitment and synchronization of brain regions during motor imagery. These findings provide important physiological evidence for understanding the neural mechanisms underlying motor-related cognition and regulation. More detailed results are illustrated in Fig. 4.

## B  ETHICS STATEMENT

This work adheres to the ICLR Code of Ethics. In this study, no human subjects or animal experimentation was involved. All datasets used, including the Visual Categorization Dataset and XP2 Dataset, were sourced in compliance with relevant usage guidelines, ensuring no violation of privacy. We have taken care to avoid any biases or discriminatory outcomes in our research process. No personally identifiable information was used, and no experiments were conducted that could raise privacy or security concerns. We are committed to maintaining transparency and integrity throughout the research process.

## C  REPRODUCIBILITY STATEMENT

We have made every effort to ensure that the results presented in this paper are reproducible. All code and datasets have been made publicly available in an anonymous repository to facilitate replication and verification. The experimental setup, including training steps, model configurations, and hardware details, is described in detail in the paper. We have also provided a full description of STBO-EC to assist others in reproducing our experiments.

Additionally, public datasets used in the paper, such as the Visual Categorization Dataset and XP2 Dataset, are publicly available, ensuring consistent and reproducible evaluation results. We believe these measures will enable other researchers to reproduce our work and further advance the field.

## D  LLM USAGE

Large Language Models (LLMs) were utilized to assist with manuscript writing and refinement. Specifically, an LLM was employed to help improve language quality, enhance readability, and ensure clarity across different sections of the paper. The model provided support for tasks including sentence restructuring, grammatical corrections, and improving the overall coherence of the text.

It should be emphasized that the LLM played no role in the conceptualization, research methodology, or experimental design of this work. All scientific concepts, research ideas, and analytical work were independently developed and executed by the authors. The LLM's contributions were strictly limited to enhancing the linguistic presentation of the paper, without any involvement in the scientific content or data analysis.

The authors assume complete responsibility for all content in the manuscript, including any text refined or generated with LLM assistance. We have carefully ensured that all LLM-assisted text complies with ethical standards and does not involve any form of plagiarism or scientific misconduct.

