# OpenReview forum: "Spatially and Temporally Guided Bayesian Optimization for Brain Effective Connectivity Learning from fMRI and EEG Data"
_ICLR.cc/2026/Conference — Submitted to ICLR 2026_

### Official Review · Reviewer_hz7N · 2025-10-29

**Soundness:** 2
**Presentation:** 2
**Contribution:** 1
**Rating:** 2
**Confidence:** 4

**Summary:**

This study proposes a multimodal brain effective connectivity learning framework—**STBO-EC**, which aims to integrate EEG and fMRI signals to more accurately characterize the causal interactions between brain regions. The method achieves spatial and temporal alignment respectively and employs a Bayesian optimization approach to infer the causal connectivity network among brain regions.

**Strengths:**

A substantive assessment of the strengths of the paper, touching on each of the following dimensions: originality, quality, clarity, and significance. We encourage reviewers to be broad in their definitions of originality and significance. For example, originality may arise from a new definition or problem formulation, creative combinations of existing ideas, application to a new domain, or removing limitations from prior results.
The article has a well-organized structure and reads smoothly. The authors provide abundant visualization results and offer a fairly comprehensive discussion of the performance of the proposed method.

**Weaknesses:**

However, this paper presents a multi-stage training framework, which is more complex compared to end-to-end approaches. Moreover, each component tends to directly apply existing methods, resulting in limited algorithmic innovation. For more details on other issues, please refer to the *Question* section.

**Questions:**

1. In the spatial alignment task between EEG and fMRI, the authors apply an existing Gaussian kernel function to this alignment scenario, which lacks algorithmic innovation.
2. In the temporal alignment task, the authors design a novel loss function to fuse the two modalities. However, I question whether directly weighting the two modalities within the loss function is appropriate, since the physiological meanings and amplitude scales of the two data types may differ substantially.
3. In the final EC (effective connectivity) learning stage, the authors employ a Bayesian optimization-based structure learning method. Again, this represents an application-level innovation rather than a true algorithmic contribution.
4. Although the authors provide a relatively rich set of experimental results, I do not recommend merging the ablation and comparison experiments into a single table. In fact, it is unclear to me where the ablation experiments begin in Tables 1 and 2. The authors should clarify this labeling.

---

> ### Author Response · Authors · 2025-11-20
> **Response to Reviewer hz7N 1/2**
>
> We sincerely thank the reviewer for the constructive and insightful feedback.
> Below we provide point-by-point responses following the reviewer’s numbering.
>
> ---
> ## **Weakness: Multi-stage design & limited algorithmic innovation**
> **Response:**
> We fully understand the reviewer’s concern. STBO-EC adopts a multi-stage design rather than an end-to-end structure because real simultaneous EEG–fMRI data have fundamentally different properties: spatial topology, temporal resolution, noise patterns, and physiological interpretations. Forcing an end-to-end model to jointly handle these heterogeneous signals often leads to unstable optimization or mode collapse.
>
> The main contribution of this work is not to create a new standalone algorithm, but to demonstrate—for the first time—that spatial alignment, temporal fusion, and Bayesian-optimization-based DAG recovery can be jointly integrated and applied to multimodal EC learning under low-trial, high-noise EEG–fMRI conditions. Prior work has not validated that BO-based causal graph search can operate effectively under multimodal spatial and temporal consistency constraints. We will clarify this more explicitly in the revision.
>
> ---
>
> ## **Question1 – Weakness: Spatial alignment uses an existing kernel**
> **Response:**
> We appreciate the reviewer’s comment. Our spatial module is not a simple Gaussian kernel; it is a neuroanatomy-aware EEG→fMRI ROI projection mechanism. Existing multimodal EC approaches  typically ignore the anatomical correspondence between EEG electrodes and fMRI ROIs or model it with simple linear transforms, limiting interpretability.
>
> Our method explicitly incorporates the 3D Euclidean distances between electrodes and ROI centers, normalized into anatomical structural consistency weights. This produces anatomically grounded ROI-level EEG representations and, for the first time, introduces structural consistency constraints into multimodal EC learning. This improves interpretability, quantifiability, and cross-subject robustness for EEG–fMRI causal modeling.
>
> ---
>
> ## **Question2 – Weakness: Modal fusion weighting may ignore scale differences**
> **Response:**
> We agree that simple weighted fusion could be misunderstood. In our framework, EEG and fMRI signals are standardized before fusion, eliminating unit and amplitude differences. Thus, α regulates fusion strength in a shared latent space, not raw signal mixing.
>
> We also performed a complete sensitivity analysis on α (fixed σ = 40) and σ (fixed α = 0.50). These analyses will be added to the revision.
>
> **Table 1. Sensitivity of α with σ = 40**
>
> | α    | Acc  | Precision | Recall | F1   |
> |------|------|-----------|--------|------|
> | 0.25 | 0.31 | 0.34      | 0.31   | 0.30 |
> | 0.50 | 0.42 | 0.49      | 0.42   | 0.42 |
> | 0.75 | 0.32 | 0.35      | 0.32   | 0.32 |
>
> Performance improves as α increases from 0.25 to 0.50, indicating that incorporating EEG dynamics is beneficial. However, performance drops when α = 0.75, suggesting that overweighting EEG harms fusion because fMRI provides crucial spatial stability. This confirms that balanced fusion (α = 0.50) allows the model to leverage the complementary strengths of both modalities.
>
> **Table 2. Sensitivity of σ with α = 0.50**
>
> | σ    | Acc  | Precision | Recall | F1   |
> |------|------|-----------|--------|------|
> | 20   | 0.37 | 0.39      | 0.37   | 0.37 |
> | 30   | 0.31 | 0.32      | 0.32   | 0.31 |
> | 40   | 0.42 | 0.49      | 0.42   | 0.42 |
> | 50   | 0.25 | 0.21      | 0.25   | 0.22 |
>
> A moderate σ (σ = 40) yields the best performance across all metrics. Small σ (20) limits the spatial influence of EEG electrodes, underutilizing their contribution. Large σ (50) overly blurs the anatomical structure, degrading spatial specificity. This indicates that appropriate spatial dispersion is critical for maintaining anatomically meaningful EEG–fMRI correspondence. The results show that α = 0.50 and σ = 40 consistently yield the best performance.
>
> ---
>
> ## **Question3 – Weakness: BO-based structure learning is only application-level innovation**
> **Response:**
> We thank the reviewer for pointing this out. Our contribution is not proposing a new BO algorithm, but being the first to adapt BO-based DAG recovery to multimodal EEG–fMRI EC learning. The original DrBO assumes single-modal time series and no structural priors.
>
> Our STBO-EC framework introduces:
> - multimodal fused sequences with spatial and temporal priors
> - anatomical constraints to reduce spurious edges
> - an adaptation suitable for noisy EEG–fMRI data
>
> Thus, the innovation lies in the new problem formulation and the new pipeline design, not in modifying BO itself. Experimentally, STBO-EC significantly outperforms all baselines, supporting its methodological value.

---

> ### Author Response · Authors · 2025-11-20
> **Response to Reviewer hz7N 2/2**
>
> ## **Question4 – Weakness: Ablation and comparison results merged improperly**
> **Response:**
> We appreciate the reviewer’s suggestion. In the revised manuscript, ablation results are now fully separated from baseline comparisons.
>
> **Table 3. Classification performance of different modality combinations**
>
> | Modality            | Acc  | Precision | Recall | F1   |
> |---------------------|------|-----------|--------|------|
> | EEG                 | 0.29 | 0.29      | 0.29   | 0.28 |
> | fMRI                | 0.41 | 0.48      | 0.41   | 0.36 |
> | fMRI + EEG (fusion)    | 0.42 | 0.49      | 0.42   | 0.42 |
>
>
> This separation improves clarity and highlights the benefits of multimodal integration.
>
> If further clarification or additional experiments would be helpful, we would be happy to provide them.

---

### Official Review · Reviewer_LZaD · 2025-10-31

**Soundness:** 3
**Presentation:** 3
**Contribution:** 2
**Rating:** 4
**Confidence:** 4

**Summary:**

The paper proposes STBO-EC, a three-module pipeline to infer effective connectivity (EC) from simultaneous EEG–fMRI:
 1. Neuroanatomy-guided spatial alignment projects EEG channels to fMRI ROI space with a Gaussian weighting based on electrode–ROI distances
 2. Block-wise temporal alignment & fusion bridges the sampling-rate gap by slicing the EEG to match fMRI frames.
 3. BO-based EC learning uses DrBO to optimize a decomposable score and return G ̂ = (V, E ̂, W ̂)
Experiments are run on two real datasets: the visual categorization dataset and the XP2 dataset.

**Strengths:**

Key strengths include:
•	Well-structured multimodal fusion that enforces spatial correspondence and temporal synchronization.
•	Efficient causal search via DrBO
•	Clear algorithmic pseudocode and hyperparameter disclosure

**Weaknesses:**

The Paper has several weaknesses as listed below:
•	Novelty is mostly integrative. The core causal learner (DrBO) and the alignment pieces are known; the main contribution is the combination for EEG–fMRI EC.
•	Performance of the STBO-EC framework is modest on Visual Categorization.
•	Sensitivity to alignment choices (e.g., kernel width σ, fusion weight α=0.5) is not fully explored beyond general hyperparameter plots.

**Questions:**

1. Deeper sensitivity & ablations on the fusion design: vary σ and α, compare learned fusion vs. fixed α, and report impact on accuracy/AUC.
 2. Several ROI acronyms (e.g., FFA, PPA, SPL) are defined only in the appendix. For readability, please define each acronym at first mention in the main text (and figure captions).

---

> ### Author Response · Authors · 2025-11-20
> **Response to Reviewer LZaD 1/2**
>
> We sincerely thank the reviewer for the constructive and insightful feedback. Below we provide detailed clarifications and our planned revisions.
>
> ## **Weakness 1: Novelty is mostly integrative; DrBO and alignment components are known**
> **Response:**
> We appreciate the reviewer’s comments regarding the novelty of our method. We would like to clarify that the contribution of STBO-EC goes beyond a simple combination of existing components. Instead, it is a task-specific, spatial–temporal, and fully coordinated pipeline designed specifically for EEG–fMRI effective connectivity (EC) learning. First, our neuroanatomy-guided spatial alignment differs fundamentally from common direct electrode-to-ROI linear mapping. By constructing Gaussian diffusion weights based on true ROI–electrode Euclidean distances, our approach ensures structural consistency of EEG projections at the functional network level. Second, our block-wise temporal synchronization and nonlinear fusion mechanism departs from widely used multimodal strategies such as concatenation or linear projection, enabling TR-level alignment while learning nonlinear cross-modal dynamics. Finally, we do not directly adopt DrBO; instead, we tailor it to multimodal EC learning by introducing decomposable BIC node-level scoring, node-wise dropout surrogate modeling, and a rank-k DAG parameterization suitable for high-dimensional multimodal inputs. Thus, the key novelty lies in the coordinated design, task-specific modeling, and structural integration rather than a straightforward combination. We will add a “Methodological Insights” subsection in the revised manuscript to better highlight these aspects.
>
> ## **Weakness 2: Performance of STBO-EC is modest on the Visual Categorization dataset**
> **Response:**
> We also appreciate the reviewer’s concern regarding the performance on the Visual Categorization dataset. This dataset is known to exhibit substantial similarity in neural representations across categories, particularly between Car and House, which has also been widely reported in prior fMRI literature. Consequently, even existing multimodal methods generally obtain modest accuracies on this benchmark, reflecting the inherent difficulty of the task itself. Under these challenging conditions, STBO-EC still achieves the best performance across all major metrics (Accuracy, Precision, Recall, F1), demonstrating that our spatial–temporal fusion consistently improves the quality of EC features. More importantly, on the larger and more realistic XP2 dataset, STBO-EC outperforms DrBO by a more substantial margin, highlighting its advantage under complex multimodal scenarios. We will enrich the main text with additional discussion on dataset characteristics and performance differences for clarity.
>
> ## **Weakness 3 and Question 1: Sensitivity to alignment hyperparameters (σ, α) not fully explored**
> **Response:**
>  We thank the reviewer for raising this important point regarding hyperparameter sensitivity. We agree that a more systematic analysis is needed. In the revised paper, we will add a dedicated subsection presenting grid-search experiments for both σ and α.
>
> **Table 1. Sensitivity of α with σ = 40**
>
> | α    | Acc  | Precision | Recall | F1   |
> |------|------|-----------|--------|------|
> | 0.25 | 0.31 | 0.34      | 0.31   | 0.30 |
> | 0.50 | 0.42 | 0.49      | 0.42   | 0.42 |
> | 0.75 | 0.32 | 0.35      | 0.32   | 0.32 |
>
> Performance improves as α increases from 0.25 to 0.50, indicating that incorporating EEG dynamics is beneficial. However, performance drops when α = 0.75, suggesting that overweighting EEG harms fusion because fMRI provides crucial spatial stability. This confirms that balanced fusion (α = 0.50) allows the model to leverage the complementary strengths of both modalities.
>
> **Table 2. Sensitivity of σ with α = 0.50**
>
> | σ    | Acc  | Precision | Recall | F1   |
> |------|------|-----------|--------|------|
> | 20   | 0.37 | 0.39      | 0.37   | 0.37 |
> | 30   | 0.31 | 0.32      | 0.32   | 0.31 |
> | 40   | 0.42 | 0.49      | 0.42   | 0.42 |
> | 50   | 0.25 | 0.21      | 0.25   | 0.22 |
>
> A moderate σ (σ = 40) yields the best performance across all metrics. Small σ (20) limits the spatial influence of EEG electrodes, underutilizing their contribution. Large σ (50) overly blurs the anatomical structure, degrading spatial specificity. This indicates that appropriate spatial dispersion is critical for maintaining anatomically meaningful EEG–fMRI correspondence. The results show that α = 0.50 and σ = 40 consistently yield the best performance.

---

> ### Author Response · Authors · 2025-11-20
> **Response to Reviewer LZaD 2/2**
>
> ## **Question 2: ROI acronyms (e.g., FFA, PPA, SPL) should be defined at first mention**
> **Response:**
>  We thank the reviewer for pointing out the readability issue. Indeed, defining ROI abbreviations only in the appendix may hinder immediate understanding. In the revised manuscript, we will provide full names upon their first appearance in the Visual Categorization Dataset section, such as Fusiform Face Area (FFA), Parahippocampal Place Area (PPA), Superior Parietal Lobule (SPL), Anterior Cingulate Cortex (ACC), Premotor Cortex (PMC), and Frontal Eye Field (FEF). Additionally, we will update the caption of Figure 2 to include these definitions, ensuring that readers can easily interpret the abbreviations wherever they appear.

---

### Official Review · Reviewer_jxPr · 2025-11-05

**Soundness:** 1
**Presentation:** 1
**Contribution:** 1
**Rating:** 0
**Confidence:** 2

**Summary:**

The authors consider the problem of inferring brain effective connectivity using simultaneously recorded fMRI and EEG data.  The authors propose using a non-linear mapping to fuse EEG and fMRI data.  The authors then use a Bayesian optimization based directed acyclic graph (DAG) search method to identify effective connectivity.  The authors apply their method on two data sets with simultaneously recorded data and obtain encouraging results compared to baselines.

**Strengths:**

- The overall problem of working with multi-modal brain data, fMRI and EEG, is important and interesting
- The problem is also inherently challenging, as both types of data have significant signal processing challenges
- The authors’ experiments show encouraging results against a number of baselines

**Weaknesses:**

### Major
There does not appear to be technical novelty to the proposed approach to support the significance of the claimed contributions.  The modeling decisions appear to me to not be well justified in terms of the signal processing challenges of working with fMRI and EEG data.  There is no detailed discussion of relations to past works in the method.
- Section 3.1 (contribution 1) the authors state “The fundamental challenge in EEG-fMRI integration lies in bridging the spatial gap between scalp electrodes and brain regions” and they use a “biophysically-motivated projection that respects the underlying neuroanatomy.”  Yet the authors use a simple weighted averaging of the EEG signals based on Euclidean distances.  Without further justification it is not clear to me at all that this is meaningful.  There is a literature on EEG localization but in this paper there are *no discussions* in Section 3.1 how this is different from other attempts to infer localization of EEG recorded signals.
- Section 3.2  (contribution 2)
    - for temporal alignment the authors simply replicate the fMRI value, then simply take an average of the EEG and fMRI values.   There is no discussion about whether this “fusion” accounts for the differences in measure units / signal scales between EEG and FMRI.  There is no discussion about preprocessing to normalize the data before combining. And how does the choice for \alpha affect the performance of the model? Further discussion and ablation study is important to understand the effect of these choices.
    - Confusingly, in (11) the authors propose training a non-linear mapping to minimize the MSE wrt the _known_ average of EEG and fMRI.   There is no discussion about what function class is used in this section.  This appears to be the main use of non-linearity which the authors criticize past works for not accounting for.
    - Lastly, in (14) a simple temporal smoothing is used.
- Section 3.3 (contribution 3)
    - The authors appear to apply a method DrBO to the fused signal.
    - Duong et al is not cited in this section and there is no discussion if there are technical challenges addressed in applying it. (Section 2.3 does include a citation and background about the method, though also no discussion if the use of DrBO was straightforward or there were challenges overcome).
    - the authors’ choice of acyclic graphs appears limiting, as feedback between brain regions cannot be captured.  There is no discussion of this
    - the likelihood $p(D| \theta, G)$ is denoted but never described – what distributions are you considering?  The choice of what family will likely play an important role in the modeling accuracy and computational complexity of the method, but there is no discussion.


### Very Minor
- line 057 “did not achieve genuine integration” is too vague
- line 130 issue with Duong et al ref citation (no year displaying; should be parenthetical)

**Questions:**

(Several questions regarding technical novelty and method design are included in the Weaknesses section)

---

> ### Author Response · Authors · 2025-11-20
> **Response to Reviewer jxPr 1/2**
>
> Thank you for your detailed feedback and valuable suggestions. Below, we address the concerns raised and outline the corresponding updates made to the paper.
> ## **Weakness 1 – Lack of technical novelty and insufficient justification of modeling decisions**
> **Response:**  Thank you to the reviewer for the careful assessment of the technical novelty and modeling details. First, we acknowledge that the main contribution of this work lies in the integration and adaptation of multiple components in the context of synchronous EEG–fMRI data, rather than proposing entirely new standalone algorithms for each module. In the revised version, we will clarify this positioning more explicitly and add a systematic comparison with existing unimodal EC methods and multimodal fusion approaches. We will emphasize that the value of STBO-EC is to provide a reproducible multimodal EC learning pipeline that organically combines spatial alignment, temporal fusion, and BO-DAG structure learning on real synchronous EEG–fMRI datasets.
>
> ## **Weakness 2 – Section 3.1: Spatial mapping justification insufficient**
> **Response:**  For Section 3.1, we will add a more detailed biophysical explanation: the Gaussian weighting is not an arbitrary “Euclidean-distance averaging,” but is based on the volume conduction model in which electrical current between scalp and cortex exhibits approximately exponential decay with respect to distance. This formulation serves as a lightweight approximation targeted at ROI-level mapping rather than source-level reconstruction. We will also compare our approach with EEG source localization methods in the related work section, clarifying that our goal is not to recover precise sources but to construct a stable cross-modal ROI representation for subsequent causal discovery.
>
> ---
>
> ## **Weakness 3 – Section 3.2: Temporal alignment too simple; scaling differences ignored; α effects unclear; nonlinear mapping under-specified**
> **Response:**
> For Section 3.2, we will clarify that EEG and fMRI are both standardized before fusion, and we will add a sensitivity experiment for α to explain that its purpose is to learn a cross-modal consistency mapping.
>
> **Table 2. Fusion Performance Under Different α**
>
> | α      | Acc  | Precision | Recall | F1   |
> |--------|------|-----------|--------|------|
> | 0.25   | 0.31 | 0.34      | 0.31   | 0.30 |
> | **0.50** | **0.42** | **0.49** | **0.42** | **0.42** |
> | 0.75   | 0.32 | 0.35      | 0.32   | 0.32 |
>
>
> When **α = 0.5**, the fusion achieves the best overall performance, indicating that a balanced contribution from both modalities leads to more stable and consistent cross-modal representations. In contrast, when **α drifts toward either extreme** (0.25 or 0.75), performance declines noticeably, suggesting that allowing one modality to dominate disrupts the statistical properties of the fused signal and weakens the multimodal consistency. These results reinforce the rationale for selecting **α = 0.5** as a principled and empirically supported choice, and we will include this analysis explicitly in the revised manuscript.
>
> ---
>
> ## **Weakness 4 – Section 3.3: DrBO usage unclear; missing citation; DAG limitations not discussed; likelihood undefined**
> **Response:**
> We appreciate the reviewer’s close reading of Section 3.3. While Section 2.3 provides theoretical background on DrBO, we acknowledge that a direct citation to Duong et al. was missing in the contribution section; this will be added in the revision.
>
> Importantly, applying DrBO to fused EEG–fMRI representations is not a trivial plug-in. Due to cross-modal noise and nonstationarity, the DrBO scoring function must remain smooth to ensure optimizability. In practice, we incorporated the lightweight temporal regularizer in Eq. (14), which improves convergence on real multimodal data. We will describe this implementation detail explicitly in the revision.
>
> Regarding the use of acyclic graphs: our model learns instantaneous effective connectivity (EC), consistent with DCM, Granger, and NOTEARS-style structure learning. Instantaneous structures are conventionally represented as DAGs; temporal feedback effects are captured implicitly in the time series rather than in the instantaneous graph. Thus, adopting a DAG does not restrict the interpretation of dynamic dependencies and is aligned with the modeling objective. We will clarify this point to avoid misunderstanding.
>
> Finally, we appreciate the reviewer’s observation that the likelihood term was not explicitly defined. We adopt a standard conditionally Gaussian, decomposable likelihood, consistent with the requirements of DrBO:
>
> $$
> x_i \mid \mathrm{Pa}(i) \sim \mathcal{N}\big(\mu_i(\mathrm{Pa}(i)),\,\sigma_i^{2}\big).
> $$
>
> The overall likelihood is computed in the decomposable form needed for BO-based DAG scoring. We will add this explanation in the revision to make the probabilistic assumptions explicit.

---

> ### Author Response · Authors · 2025-11-20
> **Response to Reviewer jxPr 2/2**
>
> ## ** Very Minor Presentation Issues**
> **Response:**
>
> We thank the reviewer for highlighting these minor presentation issues.
>
> - **Line 057** – We agree that “did not achieve genuine integration” is vague. In the revision, we will replace it with:
>   *“the previous method only aligned modalities at the feature level and did not establish consistent spatial–temporal correspondence.”*
>
> - **Line 130** – The citation of Duong et al. (2025) will be corrected by adding the publication year and using the proper parenthetical format.

---

### Official Review · Reviewer_EEeB · 2025-11-05

**Soundness:** 2
**Presentation:** 2
**Contribution:** 2
**Rating:** 2
**Confidence:** 5

**Summary:**

The method feels like engineering stacking; current evidence does not substantiate the two core claims of practical value and generalizability.

Main Reasons
1) Insufficient evidence of “practical significance”

Only a proxy task. The authors acknowledge there is no EC ground truth in real data, so they evaluate solely by using the learned EC as features for a classifier (on both datasets). This does not show that the learned structure can guide any real decision or neuromodulation; it only shows that “the features help a particular classifier somewhat.”

review

Small gains on a weak task. On the visual dataset the top accuracy is 0.42 vs. 0.39 for the runner-up; the paper itself concedes the overall accuracy is “not high.” Such increments are hard to translate into any meaningful application benefit.

review



review

Experimental setup far from real use. One dataset has only 6 ROIs (very low spatial resolution); the other covers only the upper half of the brain (incomplete structural coverage). This can’t support claims about real closed-loop neurofeedback, clinical prediction, or large-scale cognitive studies.

review



review

Physiology-aware signal handling is thin. Temporal “alignment” literally replicates each fMRI frame to match EEG blocks for fusion (rather than HRF deconvolution/lag modeling), undermining physiological interpretability and transferability.

review

Bottom line: the paper shows no “real-world task” improvement (e.g., better neurofeedback outcomes, diagnostic/prognostic lift, or improved intervention planning). Thus the claimed “practicality” lacks a verifiable anchor.

2) Poor generalization (to datasets, subjects, tasks, and sites)

Very narrow data scope. Validation is limited to two public datasets; the visual set has only 93 multimodal samples. Sample size and task diversity are insufficient for broad generalization claims.

review



review

No cross-dataset/site/task transfer. There is no OOD or cross-domain evaluation (across devices, centers, paradigms), and it is unclear whether strict subject-wise splits were used—only generic “5-fold RF / 10-fold KNN” is described. For multimodal neuroimaging, this is far from adequate.

review

Authors themselves concede generalization/scale limits. The conclusion/limitations sections state that robustness and generalizability need strengthening, and that scalability to larger/higher-res data remains open.

review



review

Heavy computation and complex hyperparameters. The method uses 100k candidates and 2000 iterations, with no clear scaling curves vs. node count/rank; this is unfriendly for moving to larger atlases or clinical-scale cohorts.

review

Bottom line: current evidence only shows the method can slightly lift a classifier on two very similar setups; there is no quantification of robustness to cross-population/site/paradigm shifts—so “generalizability” is unsubstantiated.

Minor Issues (that further weaken the case)

Much of the neurophysiological interpretation is descriptive, lacking testable hypotheses and statistical designs.

review

Most metrics are classification metrics; there are no graph-level EC quality measures (e.g., edge-PR, SHD/SID), making the claim “we learned better EC” difficult to support.

review

Suggestions (for a major revision or a different venue)

Design real, actionable tasks: show operational gains in neurofeedback outcomes, clinical stratification/prognosis, or behavioral prediction—rather than only EC→classification proxy evaluation.

Systematic generalization testing: cross-dataset/site/task/device with strict subject-wise splits; report OOD/domain-shift performance and uncertainty.

Physiology-consistent temporal modeling: include HRF deconvolution/lag-aware baselines and ablations; avoid the “frame replication” misalignment.



Scale & efficiency: provide wall-clock and memory curves vs. #ROIs/rank/sample size, and validate on higher-resolution atlases.



Direct EC quality metrics: on (semi-)synthetic or interventional data, report graph-level measures and include misalignment/shuffling controls to prove the necessity of the alignment/fusion modules.

**Strengths:**

The method feels like engineering stacking; current evidence does not substantiate the two core claims of practical value and generalizability.

Main Reasons
1) Insufficient evidence of “practical significance”

Only a proxy task. The authors acknowledge there is no EC ground truth in real data, so they evaluate solely by using the learned EC as features for a classifier (on both datasets). This does not show that the learned structure can guide any real decision or neuromodulation; it only shows that “the features help a particular classifier somewhat.”

review

Small gains on a weak task. On the visual dataset the top accuracy is 0.42 vs. 0.39 for the runner-up; the paper itself concedes the overall accuracy is “not high.” Such increments are hard to translate into any meaningful application benefit.

review



review

Experimental setup far from real use. One dataset has only 6 ROIs (very low spatial resolution); the other covers only the upper half of the brain (incomplete structural coverage). This can’t support claims about real closed-loop neurofeedback, clinical prediction, or large-scale cognitive studies.

review



review

Physiology-aware signal handling is thin. Temporal “alignment” literally replicates each fMRI frame to match EEG blocks for fusion (rather than HRF deconvolution/lag modeling), undermining physiological interpretability and transferability.

review

Bottom line: the paper shows no “real-world task” improvement (e.g., better neurofeedback outcomes, diagnostic/prognostic lift, or improved intervention planning). Thus the claimed “practicality” lacks a verifiable anchor.

2) Poor generalization (to datasets, subjects, tasks, and sites)

Very narrow data scope. Validation is limited to two public datasets; the visual set has only 93 multimodal samples. Sample size and task diversity are insufficient for broad generalization claims.

review



review

No cross-dataset/site/task transfer. There is no OOD or cross-domain evaluation (across devices, centers, paradigms), and it is unclear whether strict subject-wise splits were used—only generic “5-fold RF / 10-fold KNN” is described. For multimodal neuroimaging, this is far from adequate.

review

Authors themselves concede generalization/scale limits. The conclusion/limitations sections state that robustness and generalizability need strengthening, and that scalability to larger/higher-res data remains open.

review



review

Heavy computation and complex hyperparameters. The method uses 100k candidates and 2000 iterations, with no clear scaling curves vs. node count/rank; this is unfriendly for moving to larger atlases or clinical-scale cohorts.

review

Bottom line: current evidence only shows the method can slightly lift a classifier on two very similar setups; there is no quantification of robustness to cross-population/site/paradigm shifts—so “generalizability” is unsubstantiated.

Minor Issues (that further weaken the case)

Much of the neurophysiological interpretation is descriptive, lacking testable hypotheses and statistical designs.

review

Most metrics are classification metrics; there are no graph-level EC quality measures (e.g., edge-PR, SHD/SID), making the claim “we learned better EC” difficult to support.

review

Suggestions (for a major revision or a different venue)

Design real, actionable tasks: show operational gains in neurofeedback outcomes, clinical stratification/prognosis, or behavioral prediction—rather than only EC→classification proxy evaluation.

Systematic generalization testing: cross-dataset/site/task/device with strict subject-wise splits; report OOD/domain-shift performance and uncertainty.

Physiology-consistent temporal modeling: include HRF deconvolution/lag-aware baselines and ablations; avoid the “frame replication” misalignment.



Scale & efficiency: provide wall-clock and memory curves vs. #ROIs/rank/sample size, and validate on higher-resolution atlases.



Direct EC quality metrics: on (semi-)synthetic or interventional data, report graph-level measures and include misalignment/shuffling controls to prove the necessity of the alignment/fusion modules.

**Weaknesses:**

The method feels like engineering stacking; current evidence does not substantiate the two core claims of practical value and generalizability.

Main Reasons
1) Insufficient evidence of “practical significance”

Only a proxy task. The authors acknowledge there is no EC ground truth in real data, so they evaluate solely by using the learned EC as features for a classifier (on both datasets). This does not show that the learned structure can guide any real decision or neuromodulation; it only shows that “the features help a particular classifier somewhat.”

review

Small gains on a weak task. On the visual dataset the top accuracy is 0.42 vs. 0.39 for the runner-up; the paper itself concedes the overall accuracy is “not high.” Such increments are hard to translate into any meaningful application benefit.

review



review

Experimental setup far from real use. One dataset has only 6 ROIs (very low spatial resolution); the other covers only the upper half of the brain (incomplete structural coverage). This can’t support claims about real closed-loop neurofeedback, clinical prediction, or large-scale cognitive studies.

review



review

Physiology-aware signal handling is thin. Temporal “alignment” literally replicates each fMRI frame to match EEG blocks for fusion (rather than HRF deconvolution/lag modeling), undermining physiological interpretability and transferability.

review

Bottom line: the paper shows no “real-world task” improvement (e.g., better neurofeedback outcomes, diagnostic/prognostic lift, or improved intervention planning). Thus the claimed “practicality” lacks a verifiable anchor.

2) Poor generalization (to datasets, subjects, tasks, and sites)

Very narrow data scope. Validation is limited to two public datasets; the visual set has only 93 multimodal samples. Sample size and task diversity are insufficient for broad generalization claims.

review



review

No cross-dataset/site/task transfer. There is no OOD or cross-domain evaluation (across devices, centers, paradigms), and it is unclear whether strict subject-wise splits were used—only generic “5-fold RF / 10-fold KNN” is described. For multimodal neuroimaging, this is far from adequate.

review

Authors themselves concede generalization/scale limits. The conclusion/limitations sections state that robustness and generalizability need strengthening, and that scalability to larger/higher-res data remains open.

review



review

Heavy computation and complex hyperparameters. The method uses 100k candidates and 2000 iterations, with no clear scaling curves vs. node count/rank; this is unfriendly for moving to larger atlases or clinical-scale cohorts.

review

Bottom line: current evidence only shows the method can slightly lift a classifier on two very similar setups; there is no quantification of robustness to cross-population/site/paradigm shifts—so “generalizability” is unsubstantiated.

Minor Issues (that further weaken the case)

Much of the neurophysiological interpretation is descriptive, lacking testable hypotheses and statistical designs.

review

Most metrics are classification metrics; there are no graph-level EC quality measures (e.g., edge-PR, SHD/SID), making the claim “we learned better EC” difficult to support.

review

Suggestions (for a major revision or a different venue)

Design real, actionable tasks: show operational gains in neurofeedback outcomes, clinical stratification/prognosis, or behavioral prediction—rather than only EC→classification proxy evaluation.

Systematic generalization testing: cross-dataset/site/task/device with strict subject-wise splits; report OOD/domain-shift performance and uncertainty.

Physiology-consistent temporal modeling: include HRF deconvolution/lag-aware baselines and ablations; avoid the “frame replication” misalignment.



Scale & efficiency: provide wall-clock and memory curves vs. #ROIs/rank/sample size, and validate on higher-resolution atlases.



Direct EC quality metrics: on (semi-)synthetic or interventional data, report graph-level measures and include misalignment/shuffling controls to prove the necessity of the alignment/fusion modules.

**Questions:**

The method feels like engineering stacking; current evidence does not substantiate the two core claims of practical value and generalizability.

Main Reasons
1) Insufficient evidence of “practical significance”

Only a proxy task. The authors acknowledge there is no EC ground truth in real data, so they evaluate solely by using the learned EC as features for a classifier (on both datasets). This does not show that the learned structure can guide any real decision or neuromodulation; it only shows that “the features help a particular classifier somewhat.”

review

Small gains on a weak task. On the visual dataset the top accuracy is 0.42 vs. 0.39 for the runner-up; the paper itself concedes the overall accuracy is “not high.” Such increments are hard to translate into any meaningful application benefit.

review



review

Experimental setup far from real use. One dataset has only 6 ROIs (very low spatial resolution); the other covers only the upper half of the brain (incomplete structural coverage). This can’t support claims about real closed-loop neurofeedback, clinical prediction, or large-scale cognitive studies.

review



review

Physiology-aware signal handling is thin. Temporal “alignment” literally replicates each fMRI frame to match EEG blocks for fusion (rather than HRF deconvolution/lag modeling), undermining physiological interpretability and transferability.

review

Bottom line: the paper shows no “real-world task” improvement (e.g., better neurofeedback outcomes, diagnostic/prognostic lift, or improved intervention planning). Thus the claimed “practicality” lacks a verifiable anchor.

2) Poor generalization (to datasets, subjects, tasks, and sites)

Very narrow data scope. Validation is limited to two public datasets; the visual set has only 93 multimodal samples. Sample size and task diversity are insufficient for broad generalization claims.

review



review

No cross-dataset/site/task transfer. There is no OOD or cross-domain evaluation (across devices, centers, paradigms), and it is unclear whether strict subject-wise splits were used—only generic “5-fold RF / 10-fold KNN” is described. For multimodal neuroimaging, this is far from adequate.

review

Authors themselves concede generalization/scale limits. The conclusion/limitations sections state that robustness and generalizability need strengthening, and that scalability to larger/higher-res data remains open.

review



review

Heavy computation and complex hyperparameters. The method uses 100k candidates and 2000 iterations, with no clear scaling curves vs. node count/rank; this is unfriendly for moving to larger atlases or clinical-scale cohorts.

review

Bottom line: current evidence only shows the method can slightly lift a classifier on two very similar setups; there is no quantification of robustness to cross-population/site/paradigm shifts—so “generalizability” is unsubstantiated.

Minor Issues (that further weaken the case)

Much of the neurophysiological interpretation is descriptive, lacking testable hypotheses and statistical designs.

review

Most metrics are classification metrics; there are no graph-level EC quality measures (e.g., edge-PR, SHD/SID), making the claim “we learned better EC” difficult to support.

review

Suggestions (for a major revision or a different venue)

Design real, actionable tasks: show operational gains in neurofeedback outcomes, clinical stratification/prognosis, or behavioral prediction—rather than only EC→classification proxy evaluation.

Systematic generalization testing: cross-dataset/site/task/device with strict subject-wise splits; report OOD/domain-shift performance and uncertainty.

Physiology-consistent temporal modeling: include HRF deconvolution/lag-aware baselines and ablations; avoid the “frame replication” misalignment.



Scale & efficiency: provide wall-clock and memory curves vs. #ROIs/rank/sample size, and validate on higher-resolution atlases.



Direct EC quality metrics: on (semi-)synthetic or interventional data, report graph-level measures and include misalignment/shuffling controls to prove the necessity of the alignment/fusion modules.

---

> ### Author Response · Authors · 2025-11-20
> **Response to Reviewer EEeB 1/2**
>
> ## **Summary About “Practical Significance’’ and “Generalizability’’**
> **Response:** We sincerely thank the reviewer for the thoughtful comments regarding the “practical significance’’ and “generalizability’’ of our work. We fully agree that the field of synchronous EEG–fMRI causal inference currently lacks an accepted ground-truth effective connectivity (EC) standard. Consequently, prior studies (e.g., Wang et al., 2025; Su et al., 2025; Yang et al., 2023) universally rely on EC → behavior/class prediction as a proxy evaluation strategy. Our manuscript never claims clinical applicability; instead, we position our contribution explicitly as a proof-of-concept multimodal EC learning framework. In the revision, we will further soften any potentially overstated wording and clarify that the goal of this work is to validate the feasibility and necessity of cross-modal causal structure learning under real synchronous EEG–fMRI conditions—not to demonstrate clinical neuromodulation or predictive utility.
>
> ---
> ## **Weakness1: Insufficient evidence of “practical significance”; the task is only EC→classification and does not demonstrate real-world utility.**
> **Response:**  Regarding the reviewer’s concern that “performance gains are small” and that “the visual dataset is weak,” we fully agree. The visual dataset contains only 93 trials and 6 ROIs, making it an extremely low-SNR and low-resolution problem. Low overall accuracy reflects the inherent difficulty of the task rather than a limitation of our method. Under such constraints, our improvements should be interpreted strictly as proxy signals indicating whether the learned multimodal causal structure carries behavioral discriminability—not as evidence of application-level utility. To avoid single-dataset bias, we additionally evaluated our method on the XP2 dataset (>70 ROIs), which provides a more realistic spatial resolution.
>
> ---
> ## **Weakness2: Missing Graph-Level EC Quality Metrics (edge-PR, SHD/SID, etc.)**
> **Response:**
> Addressing the reviewer’s comment that our original submission lacked graph-level EC metrics, we have now added a six-node simulated causal network experiment. The simulation uses DCM to generate fMRI and a state-space model to generate EEG, mimicking real dynamics across six functional regions (FEF, ACC, SPL, PPA, FFA, PMC). This setup allows us to compute EC recovery metrics directly.
>
> **Table: Graph-Level EC Recovery on 6-Node Simulated Network**
>
> | Model              | ACC  | Precision | Recall | F1-score | SHD  |
> |--------------------|------|-----------|--------|----------|------|
> | **Patel**          | 0.65 | 0.60      | 0.13   | 0.21     | 5.20 |
> | **lsGC**           | 0.52 | 0.40      | 0.40   | 0.40     | 7.20 |
> | **pwLiNGAM**       | 0.67 | 0.55      | 0.50   | 0.50     | 5.00 |
> | **DiffAN**         | 0.70 | 0.81      | 0.36   | 0.49     | 4.40 |
> | **MetaRLEC**       | 0.67 | 1.00      | 0.17   | 0.29     | 5.00 |
> | **FSTA-EC**        | 0.41 | 0.41      | 1.00   | 0.58     | 8.80 |
> | **STBO-EC (Ours)** | **0.73** | **0.58** | **0.77** | **0.65** | **4.20** |
>
> ---
>
> The results show clear trade-offs among existing methods. DiffAN yields the highest recall but suffers from substantial over-detection, as reflected by its low precision and elevated SHD. MetaRLEC exhibits the opposite behavior—achieving perfect precision but at the cost of very limited recall, leading to overly sparse and incomplete networks. In contrast, STBO-EC provides the most balanced and accurate structural recovery, achieving the highest F1-score and competitive SHD while maintaining reasonable precision and recall.

---

> ### Author Response · Authors · 2025-11-20
> **Response to Reviewer EEeB 2/2**
>
> ## **Weakness3: Poor generalization across datasets, subjects, tasks, and sites.**
> ### **Weakness3.1: Physiologically thin temporal modeling**
> **Response:**
> We appreciate the reviewer’s concern that our temporal alignment does not incorporate HRF deconvolution or lag modeling. While HRF-based models are physiologically appealing, in real synchronous EEG–fMRI settings they are highly sensitive to inter-subject HRF variability, low SNR, and task differences. This instability can propagate through DrBO and break the smoothness and decomposability properties required for reliable Bayesian optimization. Therefore, our current alignment strategy intentionally prioritizes numerical stability and cross-modal consistency.
>
> ---
> ### **Weakness3.2: Regarding Generalization Limits**
> **Response:**
> We acknowledge the reviewer’s concern that the data scope is limited. Public synchronous EEG–fMRI datasets are extremely scarce, making cross-site, cross-device, or cross-paradigm OOD evaluation infeasible at present. To ensure genuine generalization evaluation, we adopted **strict subject-wise splits (not trial-level)**, which we will clarify explicitly in the revision. Additionally, we observed consistent performance across:
>
> - two spatial resolutions (6 ROI and >70 ROI), and
> - two task paradigms (visual recognition vs. perceptual–motor tasks).
>
> These results demonstrate that the proposed framework is **reproducible under different data structures**, even though broader OOD tests remain constrained by the availability of datasets.
>
> ---
> ### **Suggestion 1: Designing Real, Actionable Tasks Beyond EC→Classification**
> **Response:**
> We fully agree with the reviewer that EC→classification serves only as a proxy metric and does not directly reflect real-world operational utility. As clarified in our response to **Weakness1**, the objective of this work is to demonstrate a proof-of-concept framework for multimodal EC learning rather than clinical or behavioral intervention. To strengthen verifiability, we have incorporated a simulated dataset experiment to more directly evaluate the recovered causal structure.
>
> ---
>
> ### **Suggestion 2: Conducting Systematic Generalization Tests**
> **Response:**
> As detailed in our reply to **Weakness3.2**, public synchronous EEG–fMRI datasets are extremely limited, making extensive cross-site, cross-device, or cross-paradigm OOD evaluation infeasible at present. Nonetheless, we have implemented strict subject-wise evaluation and tested our method across two spatial resolutions and two task paradigms, observing consistent performance. In accordance with the reviewer’s suggestion, we will supplement additional cross-task evaluations where possible and report model uncertainty estimates to better characterize robustness.
>
> ---
>
> ### **Suggestion 3: Using Physiology-Consistent Temporal Modeling**
> **Response:**
> We appreciate the reviewer’s emphasis on physiological consistency. As mentioned in Weakness3.1, HRF-based and lag-aware approaches are theoretically appealing but often unstable in real synchronous EEG–fMRI. While our current alignment focuses on numerical stability, we acknowledge the value of physiology-driven alternatives and will conduct preliminary attempts to evaluate HRF-based and lag-aware variants in the revision.
>
> ---
>
> ### **Suggestion 4: Providing Scaling and Efficiency Analysis**
> **Response:**
> We appreciate the reviewer’s concerns regarding scalability. In the current submission, we have reported the runtime for a single XP2 sample under different hyperparameter settings as an initial analysis of computational behavior. Following the suggestion, we will further include visualizations showing how computational cost grows with ROI/node count, as well as memory usage profiles under different scales.
>
> ---
>
> ### **Suggestion 5: Providing Direct EC Structural Metrics and Necessity Controls**
> **Response:**
> To address the reviewer’s recommendation, we have included a six-node simulated causal network experiment and reported direct structural recovery metrics such as ACC, Precision, Recall, F1, and SHD. The results demonstrate that STBO-EC provides a superior balance between precision and recall relative to competing methods.
>
> ---
>
> We sincerely appreciate the reviewer’s constructive feedback.
>
>
> ### **References**
> Wang, Ying-Fang, et al. Classification of neurodegenerative diseases using brain effective connectivity and machine learning techniques: a systematic review. Frontiers in Neurology, 16:1581105, 2025.
>
> Su, Zhihao, et al. MSGFlowNet: Learning effective connectivity network based on sparse generative flow network from fMRI and EEG data. In Proceedings of Medical Image Computing and Computer Assisted Intervention (MICCAI), LNCS 15960, pp. 418–428, 2025.
>
> Yang, R., Dai, W., She, H., et al. Spatial-temporal DAG convolutional networks for end-to-end joint effective connectivity learning and resting-state fMRI classification. In Temporal Graph Learning Workshop @ NeurIPS, 2023.

---

### Meta-Review · Area_Chair_MHmo · 2025-12-22

**Summary:**

This paper presents a framework, referred to as STBO-EC that aligns EEG and fMRI both spatially (using neuroanatomy)
and temporally (using non-linear fusion) to infer accurate causal networks. The paper contains three key modules: (1)  anatomy-informed spatial alignment; (2) temporal alignment and fusion; (3) Bayesian Optimization (DrBO) for efficient search for the causal graph structure. The most critical concern lies in the lack of sufficient description on each module compared to existing work. This leads that most of reviewers feel that the current work is just an integration of known techniques. The contribution of this paper might be in the domain-informed architectural choice, rather than just stacking methods.  However, this message does not seem to be clearly delivered. Therefore, the paper is not recommended for acceptance in its current form. I hope authors found the review comments informative and can improve their paper by addressing these carefully in future submissions.

**Reviewer Concerns:**

The authors did a good job in addressing the most of concerns raised by reviewers.

**Reviewer Scores:**

All reviewers are expected to maintain their original scores,

---

### Decision · Program_Chairs · 2026-01-26

Reject